
# Model physics and chemistry causing intermodel disagreement within the VolMIP-Tambora Interactive Stratospheric Aerosol ensemble

Margot Clyne[1,2], Jean-Francois Lamarque[3], Michael J. Mills[3], Myriam Khodri[4], William Ball[5,6,7], Slimane Bekki[8], Sandip S. Dhomse[9], Nicolas Lebas[4], Graham Mann[9,10], Lauren Marshall[9,11], Ulrike Niemeier[12], Virginie Poulain[4], Alan Robock[13], Eugene Rozanov[5,6], Anja Schmidt[11,14], Andrea Stenke[6], Timofei Sukhodolov[5], Claudia Timmreck[12], Matthew Toohey[15,16], Fiona Tummon[6,17], Davide Zanchettin[18], Yunqian Zhu[2] and Owen B. Toon[1,2]

[1]Department of Atmospheric and Oceanic Sciences, University of Colorado, Boulder, CO, USA
[2]Laboratory for Atmospheric and Space Physics, Boulder, CO, USA
[3]National Center for Atmospheric Research, Boulder, CO, USA
[4]LOCEAN, Sorbonne Universités/UPMC/CNRS/IRD, Paris, France
[5]PMOD WRC Physical Meteorological Observatory Davos and World Radiation Center, Davos Dorf, Switzerland
[6]Institute for Atmospheric and Climate Science, ETH Zurich, Switzerland
[7]Department of Geoscience and Remote Sensing, TU Delft, The Netherlands
[8]LATMOS/IPSL, Sorbonne Université, UVSQ, CNRS, Paris, France
[9]School of Earth and Environment, University of Leeds, Leeds, U.K.
[10]National Centre for Atmospheric Science, University of Leeds, UK
[11]Department of Chemistry, University of Cambridge, UK
[12]Max Planck Institute for Meteorology, Hamburg, Germany
[13]Department of Environmental Sciences, Rutgers University, New Brunswick, NJ, USA
[14]Department of Geography, University of Cambridge, UK
[15]Institute for Space and Atmospheric Studies, University of Saskatchewan, Canada
[16]GEOMAR Helmholtz Centre for Ocean Research Kiel, Kiel, Germany
[17]Swiss Federal Office for Meteorology and Climatology MeteoSwiss, Payerne, Switzerland
[18]Department of Environmental Sciences, Informatics and Statistics, Ca'Foscari University of Venice, Mestre, Italy

*Correspondence to*: Margot Clyne (Margot.Clyne@colorado.edu)

**Abstract.** As part of the Model Intercomparison Project on the climatic response to Volcanic forcing (VolMIP), several climate modeling centers performed a coordinated pre-study experiment with interactive stratospheric aerosol models simulating the volcanic aerosol cloud from an eruption resembling the 1815 Mt Tambora eruption (VolMIP-Tambora ISA ensemble). The pre-study provided the ancillary ability to assess intermodel diversity in the radiative forcing for a large stratospheric-injecting equatorial eruption when the volcanic aerosol cloud is simulated interactively. An initial analysis of the VolMIP-Tambora ISA ensemble showed large disparities between models in the stratospheric global mean aerosol optical depth (AOD). In this study, we now show that stratospheric global mean AOD differences among the participating models are primarily due to differences in aerosol size, which we track here by effective radius. We identify specific physical and chemical processes that are missing in some models and/or parameterized differently between models, which are together causing the differences in effective radius. In particular, our analysis indicates that interactively tracking hydroxyl radical (OH) chemistry following a large volcanic injection of sulfur dioxide ($SO_2$) is an important factor in allowing for the timescale for sulfate formation to be properly simulated. In addition, depending on the timescale of sulfate formation, there can be a large difference in effective radius and subsequently AOD that results from whether the $SO_2$ is injected in a single model gridcell near the location of the volcanic eruption, or whether it is injected as a longitudinally averaged band around the Earth.



## 1 Introduction

The Model Intercomparison Project on the climatic response to Volcanic forcing (VolMIP) devised a co-ordinated multi-model experiment to assess the volcanic aerosol cloud from a large equatorial stratospheric-injecting eruption, as simulated by state-of-the-art climate models with interactive stratospheric aerosols (the VolMIP-Tambora ISA ensemble). The original goal of the Tambora ISA ensemble was to define a consensus forcing dataset that would be used for the VolMIP *volc-long-eq* experiment, which provides a reference aerosol dataset to effect a common volcanic forcing in simulations of the climate response to an eruption

similar to 1815 Mt. Tambora (Zanchettin et al, 2016). The climate models running the VolMIP *volc-long-eq* experiment will not simulate the volcanic aerosol cloud interactively, the experiment designed to ensure all models specify the same reference aerosol optical properties for the volcanic forcing. The VolMIP-Tambora ISA ensemble experiment is similar in approach to the ongoing Interactive Stratospheric Aerosol Model Intercomparison Project's (ISA-MIP)'s Historical Eruptions $SO_2$ Emission Assessment (HErSEA) experiment (Timmreck et al., 2018), which intercompares model simulations of the three largest major eruptions of the

20th century, but for the HErSEA experiment, the models run different realizations of the volcanic aerosol cloud, based on a small number of alternative specified $SO_2$ emission and injection heights for each eruption. In the VolMIP-Tambora ISA ensemble experiment, climatological variables and injection parameters were prescribed under a coordinated experimental protocol embedding historical information about the 1815 Mt. Tambora eruption to reduce intermodel differences from initial conditions. The experimental protocol designated an emission of 60 Tg of sulfur dioxide ($SO_2$) into the stratosphere, approximately 2.6 to 4.3

times the emission estimates for the 1991 Mt. Pinatubo eruption (Carn et al., 2016). An initial assessment of the VolMIP-Tambora ISA ensemble carried out by Zanchettin et al. (2016) showed substantial differences among the participating model's predictions for the Tambora cloud's global dispersal, in particular, between the timing and magnitude of the peak global mean stratospheric aerosol optical depth (AOD).

As it was intended to be a relatively straightforward experiment, the large spread in model outputs surprised the VolMIP community (Khodri et al., 2016, Zanchettin et al., 2016). After fixing errors found in the implementation of the injection protocol in some of the models, subsequently updated simulations (which are used here and in Marshall et al., 2018) from the participating models produce intermodel disagreement of stratospheric global mean AOD that is just as drastic (Figure 1). These disparities, and a lack of understanding of their origin, led to a decision not to use the VolMIP-Tambora ISA ensemble to generate the consensus

dataset of aerosol optical properties to be used as volcanic forcing input for the VolMIP *volc-long-eq* experiment, as was originally intended (Zanchettin et al., 2016). Instead, the input volcanic forcing of aerosol optical properties was taken from the Easy Volcanic Aerosol (EVA) forcing generator (Toohey et al., 2016). The EVA forcing generator is based on analytical functions, and does not simulate microphysical processes. However, due to the large differences in results with the aerosol models, the causes of which were not understood at the time, EVA was elected as a more idealized but more understandable reference forcing.


Marshall et al. (2018) also analysed the VolMIP-Tambora ISA ensemble, finding significant intermodel differences in the timing, magnitude and spatial patterns of the volcanic sulfate deposition to the Greenland and Antarctic ice sheets. For example, the analysis showed that the ratio of hemispheric peak atmospheric sulfate aerosol burden after the eruption to the average ice-sheet-deposited sulfate varies between models by up to a factor of 15. The study suggested general reasons for the intermodel

disagreement in sulfate deposition to be MAECHAM5-HAM's use of prescribed OH, intermodel differences in simulated stratospheric aerosol transport that are in part due to simulated stratospheric winds and horizontal model resolution, and differences in stratosphere-troposphere exchange of aerosol that are in part due to different deposition and sedimentation schemes and vertical model resolution.



The LMDZ-S3A model was not added to the VolMIP-Tambora ISA ensemble until recently, after the Marshall et al. (2018) paper
was published. In this paper we go further than Marshall et al. by pinpointing the primary sources of intermodel inconsistencies in
volcanic aerosol formation, evolution and duration in the stratosphere that largely contribute to the inconsistencies in modeled
global stratospheric AOD. We explain where and why these specific differences matter for AOD. We illustrate how the sources of
disagreement in AOD that we identify in this paper, most crucially those relating to aerosol particle size whose importance was
not analyzed in Marshall et al. (2018), also apply to volcanic sulfate deposition. We end by providing possible ways to move
forward to address these uncertainties in future intercomparison studies.

**2 Methods**

The protocol for the VolMIP-Tambora ISA ensemble (Table 5 of Zanchettin et al., 2016) called for an equatorial injection of 60
Tg of $SO_2$ (equivalent to ~30 TgS) on April $1^{st}$, 1815 for a 24-hour eruption with 100% of the mass injected between 22 and 26
km, increasing linearly with height from zero at 22 km to max at 24 km, and then decreasing linearly to zero at 26 km. Modeling
groups injected at the nearest corresponding vertical levels available on their model vertical grid. This $SO_2$ emission estimate is
roughly in agreement with prior petrological and ice core estimates (e.g., Self et al., 2004; Gao et al., 2008). The 60 Tg injection
also agrees with the subsequent estimate of Toohey and Sigl (2017), who provide an uncertainty estimate of ± 9 Tg $SO_2$ (4.5 TgS).
Further explanation about the decision of the injection parameter values used for the experimental protocol can be found in Marshall
et al. (2018). Ensembles with five members were run for five years producing monthly average outputs, and were started at the
easterly phase of the quasi-biennial oscillation (QBO). Radiative forcings for $CO_2$, other greenhouse gases, tropospheric aerosols
(and $O_3$ if specified in the model), were set to the values each model uses to define preindustrial (1850) climate conditions. In the
Community Earth System Model – Whole Atmosphere Community Climate Model (CESM-WACCM), the simulations were run
with a preindustrial coupled atmosphere and ocean. In the Laboratoire de Météorologie Dynamique zoom - Sectional Stratospheric
Sulfate Aerosol (LMDZ-S3A) model, and ECHAM-HAM in the Middle Atmosphere version (MAECHAM5-HAM), and the
modeling tool for studies of SOlar Climate Ozone Links- Atmospheric and Environmental Research (SOCOL-AER), and the
Unified Model – United Kingdom Chemistry and Aerosol (UM–UKCA), simulations did not include interactive coupling between
atmosphere and ocean, but instead were run with prescribed sea-surface temperatures from a previous coupled atmosphere-ocean
pre-industrial control integration.

Some characteristics of the VolMIP-Tambora ISA models are included in Table 1. One important difference between the
simulations is how some of the modelling groups included additional runs with an artificial longitudinal spread of the volcanic
cloud. The cloud from an equatorial injection of this size into the stratosphere will fully encircle the globe within the tropics in a
few weeks, spreading (in this case) westward with the zonal winds from the easterly phase of the QBO (Robock and Matson, 1983;
Baldwin et al., 2001). To investigate the potential impact of beginning with a 2-D zonal injection of $SO_2$ instead of a 3-D injection
that incorporates longitude as a dimension, the MAECHAM5-HAM and SOCOL-AER modeling groups performed both "point"
and "band" experiments. We refer to a "point" injection as a grid cell at the equator at the longitude of Tambora, which is located
at 118°E, and a "band" injection as a zonal injection of the 60 Tg of $SO_2$ spread evenly across all longitudes at the grid latitude
nearest to the equator. CESM-WACCM and UM-UKCA injected the 60 Tg of $SO_2$ as point injections. LMDZ-S3A performed a
band injection. As a 2-D scaling based forcing generator, EVA does not follow the injection from its origins for stratospheric
transport, and instead uses a three-box model to produce the zonally-averaged spatiotemporal structure of the cloud. In EVA, $SO_2$





is converted to sulfate based on a fixed timescale, and effective radius is taken to be proportional with aerosol mass following the observed effective radius evolution after Pinatubo. EVA does not take into account the stratospheric sulfur injection height, nor does it account for vertical variations in stratospheric dynamics (Toohey et al., 2016). The term "VolMIP-Tambora ISA ensemble

mean" refers to the average of all models except for the MAECHAM5-HAM band and SOCOL-AER band injection experiments to avoid double counting of the same model with its point injection experiment. The post-processing methods to obtain the monthly stratospheric global mean values of AOD, sulfur species burdens, and effective radius are detailed in Appendix A. $e$-folding lifetimes are calculated as the time elapsed after reaching the maximum value when the quantity crosses $1/e$ of its maximum. The precision of these $e$-folding rates is limited by the time resolution of the results, which are output every month.


The requested wavelength from the experiment protocol for AOD in the visible was $\lambda = 525$ nm, but the actual wavelengths at which models provided their outputs are presented in Table 1. While different wavelengths were used within the visible band, they still fall within the Mie scattering regime for volcanic sulfate aerosols, because the optical size parameter of $\alpha = \frac{2\pi r}{\lambda}$ remains within the order of 1-10 for particles of radius 0.1-1 μm. Specifically, SOCOL-AER AOD reported values over a wide wavelength range,

as shown in Table 1. However, SOCOL-AER produces relatively large particles, as discussed below, with size parameters around seven for 525 nm wavelength. In the large particle limit for optical scattering, AOD is not very wavelength dependent, so SOCOL-AER's wide wavelength range is unimportant when comparing peak AOD magnitudes. SOCOL-AER and LMDZ-S3A use sectional size distribution schemes. The rest of the models use modal size distribution schemes. Further details about the size distribution schemes used by the models can be found in Table 2 and Appendix B.


UM-UKCA produces an internally generated QBO (Table 2) so each of its five runs has a slightly different QBO strength even though they all inject the volcanic $SO_2$ with an easterly phase in the six months after the injection. In LMDZ-S3A, winds and temperatures are nudged towards ERA-Interim reanalyses, treating the Tambora period as the Mt. Pinatubo period, which begins during the easterly phase of the QBO (i.e. starting with 1991 being 1815 and so on). SOCOL-AER and CESM-WACCM nudge

the QBO to be in the easterly phase at the time of injection by nudging the winds in the tropics to historical observations. SOCOL-AER uses the QBO strength observed during and after 1991 Mt. Pinatubo. Three of the ensemble runs in CESM-WACCM use the QBO observed after Mt. Pinatubo starting in 1991, and two CESM-WACCM ensemble runs use the QBO strength observed after El Chichón starting in 1982. MAECHAM5-HAM does not generate a QBO at the resolution used here: equatorial winds are persistently easterly. EVA does not account for the QBO in its transport scheme.


After $SO_2$ is injected in the manner described by the experimental protocol, it is converted to $H_2SO_4$ gas (sulfuric acid vapor) with the rate-limiting step being the reaction with photochemically produced OH (Bekki, 1995). The strong volcanic source of $H_2SO_4$ gas nucleates to produce an aerosol cloud that initially comprises very small particles (a few tens of nm). These then rapidly coagulate with each other and grow also from acid vapor condensation, to submicron sized particles (English et al., 2011; Seinfeld

and Pandis, 2016). In this paper we write the particle form of $H_2SO_4$ as "$SO_4$" to distinguish between the vapor phase and the particle phase. Sulfate aerosol ($SO_4$) is the species of sulfur directly relevant to AOD. More detailed descriptions of the sulfur chemistry can be found in the model overview references cited in Table 1. The stratospheric residence time of the sulfate is controlled by advective transport, which is independent of particle size, and by vertical fall velocity, which depends on particle size. In Sect. 3.1-3.3, we provide an overview of the results from the different models, focusing on the global mean values of

stratospheric AOD, sulfate burden, and effective radius.





MAECHAM5-HAM and LMDZ-S3A do not interactively calculate OH, and instead prescribe OH concentrations (Table 2). In LMDZ-S3A, the OH fields give a stratospheric mean lifetime of about 36 days for $SO_2$. Because it was not included in the injection experimental protocol, none of the models considered an injection of water, which could impact the OH mixing ratios, or ash which could be important for photolysis (Sect. 4.4). The impact of band injections and OH chemistry on AOD, sulfate burden, and effective radius are discussed in Sect. 4.2.

# 3 Results

## 3.1 Global-mean stratospheric AOD

Ensemble means of global mean stratospheric AOD outputs from participating models are plotted in Figure 1. They are wide-ranging both in magnitude and time. For global mean stratospheric AOD; the peak values of the models vary by 65% above to 19% below the multi-model mean maximum value for the original VolMIP-Tambora ISA ensemble models that were included in Marshall et al. (2018), and the peak values vary by 63% above and 34% below the multi-model mean maximum when LMDZ-S3A is included. The model outputs with higher than average AOD are CESM-WACCM, MAECHAM5-HAM band, and UM-UKCA. We will refer to this group as "Group AODHigh". The model outputs with lower than average AOD ("Group AODLow") are MAECHAM5-HAM point, EVA, SOCOL-AER point, SOCOL-AER band, and LMDZ-S3A band. The mean AOD values for Group AODHigh and Group AODLow for the first year after the injection (April 1815 – March 1816) are 0.49 and 0.28 respectively. The ensemble mean AOD lies between these two subsets, and is 0.36 for the first year.

The injection of $SO_2$ occurred on April 1st 1815. The LMDZ-S3A band and MAECHAM5-HAM band injections reach their peak AOD in July 1815, with values of 0.27 and 0.61 respectively. MAECHAM5-HAM point and UM-UKCA peak a month later with AOD values of 0.36 and 0.53 respectively. SOCOL-AER point, SOCOL-AER band, and EVA peak at 0.37, 0.36, and 0.35 in December 1815, and CESM-WACCM finally peaks at 0.67 in April 1816, a full year after the injection. While MAECHAM5-HAM band and CESM-WACCM are the two models that reached the highest magnitudes for stratospheric global mean AOD, CESM-WACCM remains at AOD levels above an arbitrary value of 0.1 for almost a year and a half longer than does MAECHAM5-HAM band (38 vs. 21 months). Once AOD begins to decline, CESM-WACCM and UM-UKCA have AOD *e*-folding times of 17 months, EVA is 15 months, SOCOL-AER point, SOCOL-AER band, and MAECHAM5-HAM band are 11 months, and MAECHAM5-point and LMDZ-S3A band are 10 months (Table 3). Interestingly, the band injection for MAECHAM5-HAM produces twice the peak AOD of its point injection. However, within the SOCOL-AER runs there is little difference in AOD between the band and point experiments. We have detailed discussions on band and point injections in Sect. 4.2.2.

## 3.2 Stratospheric sulfate burden

We split the following description of the results on stratospheric sulfate burden into two parts. In Section 3.2.1 we present the results without including LMDZ-S3A and then we separately explain the LMDZ-S3A results in Sect. 3.2.2. This is because the LMDZ-S3A sulfate burden results are very different from the other models, and we do not want an analysis of the variation between the remaining models to be overwhelmed by discussion about the differences of a single model.



### 3.2.1 Stratospheric sulfate burden without LMDZ-S3A

Mass of global stratospheric sulfur is conserved in the models with sulfur aerosol chemistry within the first several months following the injection of $SO_2$, as the sums of their volcanic sulfur species burdens ($SO_2 + H_2SO_4 + SO_4$) stabilize at ~30 TgS, but then decay at different rates (Supplemental Figure S1). The relevant form of volcanic sulfur for AOD is sulfate aerosol ($SO_4$),

whose global stratospheric burden time series (in TgS) is shown in Figure 2. All of the models produce peak sulfate global burdens of 27-29 TgS, but these peak values are reached at different times, and sulfate is removed from the global stratosphere at different rates.

Table 3 provides more insight on the sulfate burden. All models peak in $SO_2$ burden at the first month of the experiment, which is

in April 1815. Model outputs are provided monthly, so some $SO_2$ has already been removed or converted by the time of the first month's output. MAECHAM5-HAM gives the quickest conversion time from $SO_2$ to sulfate, as indicated by the short (<1 month) $e$-folding stratospheric lifetime of $SO_2$ and by the earliest peak in sulfate, which occurs in August 1815.

In Table 3 we see that MAECHAM5-HAM produces the shortest duration of elevated sulfate in the stratosphere, with an $e$-folding

time of 8 months for the point injection, and 10 months for the band injection. Sulfate burdens of the other models continue to rise after the MAECHAM5-HAM sulfate burden has already begun to decrease. With a longer $SO_2$ $e$-folding time of 2 months, SOCOL-AER reaches its peak sulfate burden in November 1815, after which sulfate is removed at the same rate as in MAECHAM5-HAM's band injection. Figure 2 indicates that UM-UKCA and CESM-WACCM have more stabilized elevated sulfate burdens. Both models give 2 month $e$-folding times for $SO_2$. Unlike MAECHAM5-HAM and SOCOL-AER, whose sulfate burdens rapidly

increase until reaching a peak value, the sulfate burdens of UM-UKCA and CESM-WACCM begin to plateau roughly 4-5 months after the injection and then increase more gradually before finally reaching their peak values in October 1815 for UM-UKCA and March 1816 for CESM-WACCM. The decay rate of the sulfate burden that follows is 4 months longer in UM-UKCA than in MAECHAM5-HAM band and SOCOL-AER. In addition to taking the longest time to reach its peak sulfate burden value, CESM-WACCM has the longest duration of elevated sulfate burden, with an $e$-folding time twice that of MAECHAM5-HAM point.

Marshall et al. (2018) find that 35% of the global sulfate deposition in MAECHAM5-HAM point occurs in 1815, and 60% occurs in 1816. In SOCOL-AER deposition starts after MAECHAM5-HAM and 75% of global sulfate deposition occurs in 1816. Only 9% occurs in UM-UKCA during 1815, and then 55% in 1816 and 29% in 1817. No sulfate deposition occurs in CESM-WACCM until 1816, when 35% of global sulfate deposition occurs followed by 46% in 1817 and 17% in 1818, with deposition levels still elevated at the end of the simulation (Marshall et al., 2018). Although they vary widely on the timing and duration of elevated

global stratospheric sulfate burden, none of the models predict that any more than 4 TgS is removed before the peak value of global stratospheric sulfate is reached. In other words, models find that more than 85% of the emitted sulfur in the volcanic $SO_2$ is converted into stratospheric sulfate aerosol.

A possible reason that the sulfate burden in CESM-WACCM continues to increase for a full year following the eruption is that it

has the highest top of all of the models, well above the mesopause (Table 1), allowing the most complete representation of the middle atmosphere circulation. UM-UKCA is the only other model to include the entire mesosphere, and is the only other model to produce a prolonged peak in the sulfur burden (lasting nearly a year), with a stratospheric lifetime second in length only to CESM-WACCM.



### 3.2.2 Stratospheric sulfate burden of LMDZ-S3A

The global stratospheric sulfate burden is noticeably lower in LMDZ-S3A than in all of the other models (Figure 2), and reaches a maximum of only 23 TgS in the band injection. Unlike the other models, the mass of global stratospheric sulfur in LMDZ-S3A is not stable within the first several months following the injection of $SO_2$; sulfur is crossing from the stratosphere into the troposphere. The sum of the volcanic sulfur species stratospheric burden ($SO_2 + H_2SO_4 + SO_4$) exceeds 29 TgS for the first two months in the LMDZ-S3A band injection experiment (April and May 1815), but then quickly drops (Figure S1). The stratospheric

$SO_2$ $e$-folding time in LMDZ-S3A of about one month is longer than MAECHAM5-HAM's, but less than CESM-WACCM, UM-UKCA, and SOCOL-AER's (Table 3). By peaking in July 1815, the LMDZ-S3A band injection has the earliest sulfate peak of all models.

### 3.3 Stratospheric effective radius

Sulfate aerosol particles continue to increase in size by condensational growth and coagulation after they are produced. The time

series of the global stratospheric mean effective radius (*Reff*) defined by Eq. (A3) is shown in Figure 3. CESM-WACCM and UM-UKCA produce the smallest *Reff*, with values never exceeding 0.5 µm. The LMDZ-S3A band injection reaches a maximum *Reff* of 0.63 µm. SOCOL-AER has larger *Reff* than the multi-model mean, with both band and point injection experiments identically peaking at 0.65 µm. The MAECHAM5-HAM point injection grows larger particles than its corresponding band injection, reaching *Reff*s of 0.73 and 0.6 µm respectively. EVA has the largest *Reff*, reaching a peak of 0.8 µm.


Despite the fact that EVA and MAECHAM5-HAM point have the largest particle sizes over a global stratospheric average (Figure 3), LMDZ-S3A produces the particles with the largest effective radius locally. Vertical profiles of effective radius in the tropics (Figure 4) show large (greater than 1 µm effective radius) particles being produced in LMDZ-S3A and already crossing the tropopause within the first month. *Reff* is calculated from the mean size of the particles that are present in the stratosphere. Details

on the calculation of *Reff* are in Appendix A. The global stratospheric mean effective radius (*Reff*) decreases with time after reaching its maximum because the larger of the sulfate aerosol particles are falling out of the stratosphere. *Reff* decreases most quickly in the simulations with the largest effective radii. LMDZ-S3A begins to decrease first, and the MAECHAM5-HAM point injection *Reff* then declines at the most accelerated rate (Figure 3). EVA has the largest *Reff* of all the models, but as mentioned earlier in Sect. 2 and described further in Appendix B, EVA assumes a particle size enacted from a mass-based scaling from the

*Reff* enhancement observed after Pinatubo. The EVA, SOCOL-AER, and MAECHAM5-HAM band experiments all decline in *Reff* at roughly the same rate after the maximum is reached, with UM-UKCA declining in *Reff* at a slightly slower rate. *Reff* in CESM-WACCM declines the most slowly out of all of the models, and is still greater than 0.3 µm by the fourth year of the simulation (Figure 3).

### 4 Discussion

This VolMIP-Tambora ISA ensemble study of an idealized equatorial large stratospheric injection of $SO_2$ based on the 1815 eruption of Mt. Tambora provides insight to significant gaps between models. These gaps are not random, nor related to small details in differences between models. Rather they are related to first order differences in the physics and chemistry in the models (to be further described in the following sections). One could argue that one should not derive a volcanic forcing parameter for global aerosol optical depth by averaging models which lack important physics with those that have more complete physics, particularly when the impacts of those simplifications are not understood. While the Marshall et al. (2018) study includes a

particularly when the impacts of those simplifications are not understood. While the Marshall et al. (2018) study includes a





comparison of the model results to observations of the 1815 Mt Tambora ice core sulfate deposits, conclusions on model performance should not be drawn based on which model or models within this VolMIP-Tambora ISA ensemble best simulate impacts from the eruption compared to observations because there are large uncertainties for the actual volcanic injection parameters. In addition, this experiment does not include volcanic injections of water or ash, which can impact the volcanic forcing.

This VolMIP-Tambora ISA ensemble uses a single prescribed set of injection parameters, which prevents individual models from choosing their injection parameters to make their results match a desired set of observations. As an idealized experiment, this study serves best to compare models with models. The goal of this paper is to understand the reasons for the intermodel disagreement in both magnitude and timescale of stratospheric global AOD shown in Figure 1.

**4.1 Key output variables defining AOD magnitude**

The simulated values of AOD and *Reff* show that global stratospheric average AOD is proportional to its aerosol mass burden divided by effective radius. Equation (1), which is adapted from Seinfeld and Pandis, (2016), describes this relationship.

$$AOD = \psi * \frac{M}{Reff} \tag{1}$$

where $M$ is the global stratospheric mass burden of sulfate in TgS, which is the quantity plotted in Figure 2. The proportionality

scalar, $\psi$, is:

$$\psi = \frac{3q}{4\rho A} \frac{(molec.\ weight\ H_2SO_4)}{(molec.\ weight\ S) * \omega} \tag{2}$$

Here $A$ is the surface area of the Earth, $\rho$ is the volume density of a sulfate aerosol particle ($H_2O$-$H_2SO_4$) in units of grams of aerosol per volume, the molecular weight of $H_2SO_4$ = 98.079 g mol$^{-1}$, the molecular weight of S = 32.065 g mol$^{-1}$, $\omega$ is the mass fraction of sulfuric acid within the $H_2O$-$H_2SO_4$ aerosol droplet, and $q$ is the extinction efficiency, which is a unitless function of

the ratio of effective radius to wavelength, and the optical constants of sulfuric acid water solutions (of which the refractive index changes with $\omega$). Equations (1 and 2) are basically exact for spheres in the limit in which the particles are all the same size, and uniformly distributed over the planet. The purpose of Eqs. (1 and 2) is to develop a simple analysis method to understand why the various models differ so much in computed AOD. The climate models are very complex, but the underlying physics relating the computed parameters of mass, optical depth and effective radius is relatively simple. A derivation of how Eqs. (1 and 2) are adapted

from the expressions in Seinfeld and Pandis (2016) is provided in the supplementary info of this paper. Evidence that this simplified model for global stratospheric AOD works is presented in Sect. 4.1.1.

In Eq. (2), $\omega$ is present because we are tracking the mass of sulfate in the models, but the particles also contain water, which makes them larger. The density is present because the optics depend on the physical size of the particles rather than their mass. There is

a large vertical gradient in $\rho$ and $\omega$ in the stratosphere due to the variation of the absolute amount of water with altitude. As particles fall from the initial injection altitude near 26 km to the tropopause, they pick up water due to the increasing amount of water vapor, making them less concentrated, but they also become less dense. Both changes make the particles larger as they drift downward. An example showing the variation of $\rho$ and $\omega$ with altitude is in the supplementary info of this paper (Figure S2).

Global stratospheric mean AOD vs. global stratospheric sulfate burden ($M$) is shown in Figure 5. Within each model, larger sulfate burden leads to higher AOD, which is as expected from Eq. (1). If $\rho$ and $\omega$ were constant and *Reff* was fixed, AOD would be a linear function of sulfate burden. However, Figure 5 shows that within the same model, AOD values can vary by up to ~0.1 for the





same sulfate burden before and after the month of peak AOD. This AOD variation is because *Reff* changes with time, and $\rho$ and $\omega$ are varying with the altitude of the cloud. As sulfate burden increases, intermodel spread of AOD grows. When the global

stratospheric sulfate burden is greater than 25 TgS different models give corresponding AOD values ranging widely from 0.34 to 0.63 (Table 4). LMDZ-S3A never reaches a global stratospheric sulfate burden of 25 TgS (Table 3). The particles in LMDZ-S3A grow large quickly and fall out of the stratosphere (Figure 4) too early to reach a global sulfate burden nearing those of the other models (Figure 2). It is unclear at this time why the particles in LMDZ-S3A grow so large in this experiment. Our hypothesis is that the particles in LMDZ-S3A are growing so large because of the equations for nucleation rates used in the model, which,

compared to some of the equations used by the other models, leads to lower nucleation rates (Appendix C).

Larger *Reff* corresponds to lower AOD (Eq. 1). In the applicable visible wavelength of 550 nm, the value of *q/Reff* decreases as effective radius increases above 0.3 µm (Figure S3). Global stratospheric mean AOD vs. effective radius is shown in Figure 6. Circles outlined in black indicate the months for each model at which the global burden of sulfate exceeds 25 TgS. During this

period, the mean effective radius of Group AODHigh (CESM-WACCM, UM-UKCA, MAECHAM5-HAM band) is 0.50 µm, with a mean AOD of 0.57. The mean effective radius of Group AODLow without EVA or LMDZ-S3A (SOCOL-AER point, SOCOL-AER band, MAECHAM5-HAM point) is 0.66 µm with a mean AOD of 0.35.

VolMIP-Tambora ISA ensemble models agree relatively well on sulfate burden during the first year after the injection, especially

toward the end of 1815, but largely disagree on AOD. If LMDZ-S3A is excluded, the spread of the peak global mean stratospheric values from individual models is only 8% above to 1% below the multi-model mean maximum for sulfate burden, vs 56% above to 19%, below the multi-model mean maximum for AOD. Figure 5 emphasizes the disagreement between models on global AOD values for a given sulfate burden. Therefore, *Reff*, which is the other major component of the AOD equation, Eq. (1), must be a key contributing factor to this intermodel disagreement during the first year after the injection. The peak *Reff* values from the

individual models vary by 27% above to 20% below the maximum value of the multi-model mean. When LMDZ-S3A is included these values change to 12% above to 11% below for sulfate burden, to 63% above to 34% below for AOD, and remain the same for *Reff*. The time series showing how the models differ on AOD, sulfate burden, and *Reff* is plotted in Figure 7. The plots of normalized intermodel variance show that during the first year after the eruption (April 1815 – March 1816), intermodel variance of AOD is primarily due to variance of *Reff*. When models agree on sulfate burden, they disagree on *Reff*. The early spike in sulfate

burden variance in Figure 7b is due to the earlier production of sulfate in MAECHAM5-HAM (discussed in Sect. 4.2.1). After the first year after the injection (i.e., after roughly March 1816), intermodel disagreement in AOD is primarily due to differences in the simulated sulfate burden. This narrative is seen more clearly via the dashed lines in Figure 7, where LMDZ-S3A is excluded from the intermodel variance, and the remaining models have a brief period when they intersect in global stratospheric sulfate burden around October 1815. In LMDZ-S3A, much of the sulfur falls out of the stratosphere early in the experiment due to the

higher falling velocity of the large particles that are produced in the model. The sulfate burden in LMDZ-S3A that remains in the stratosphere is much lower than the other models, which additionally contributes to the intermodel variance of the sulfate burden and the AOD (solid line in Figure 7).

In this experiment, Group AODHigh all yield smaller *Reff,* so the aerosol particles which they produce are more optically efficient

at scattering light. As a result, they all have higher AOD values when the sulfate burden is the same for all models (Figure 5, Figure 6) than do Group AODLow. This explains the spread in AOD magnitudes of Figure 1, versus the proximity of sulfate burden magnitudes along the same timeline in Figure 2. For example, the UM-UKCA and SOCOL-AER models differ in magnitude by





~1.4x for AOD and for *Reff*, but they have similar sulfate burdens and closely matching rates of rise and decay of AOD and *Reff*. CESM-WACCM and UM-UKCA aerosols never grow past a global stratospheric mean effective radius of 0.5 μm, which
contributes to their longer *e*-folding times for sulfate burden and AOD compared to the other models. The sulfate burden *e*-folding time is longer because smaller particles will not sediment as quickly as larger particles.

### 4.1.1 Comparing model results of AOD to AOD reconstructed from Eqs. (1 and 2)

Operationally, $\omega$ is the only unknown value when reconstructing AOD using Eqs. (1 and 2) for the VolMIP models. Values of *M* and *Reff* are known outputs from the VolMIP models. Values of *q* are calculated by Mie theory using inputs of effective radius,
wavelength set at 550 nm, and $\omega$ (to determine the complex refractive index of the aerosol). The global stratospheric average values of *q* are then calculated in the same weighted average method as is done for *Reff* in Eq. (A3). Myhre et al. (2003) show that $\rho$ can be calculated using a polynomial expansion equation with inputs of $\omega$ and temperature. In the applicable temperature range for the stratosphere and locations of the volcanic aerosol, $\rho$ is primarily a function of $\omega$. Plots of reconstructed global stratospheric average AOD using Eqs. (1 and 2) are shown by the shaded regions in Figure 9. The actual $\omega$ values were not output by the VolMIP models
at the time, so the reconstructions in Figure 9 were instead made using a single value for $\omega$ prescribed throughout the stratosphere. The shading in Figure 9 for each VolMIP model encompasses the reconstructed AOD calculated using $\omega$ ranging from 0.9 (lower edge of the shading) to $\omega = 0.75$ (upper edge of the shading). For comparison, the actual AOD from the VolMIP models (i.e. the AOD in Figure 1) is plotted as the dashed lines in Figure 9. CESM-WACCM and SOCOL-AER follow the lower part of the shaded region in the first few months, and then the upper part later. This behavior is consistent with the bulk of the aerosols having a high
weight of sulfuric acid percent initially, and then a lower weight percent as they fall downward into air with higher water concentration. Even with using global stratospheric average values for *M, Reff, q*, and (prescribed) $\omega$, Eqs. (1 and 2) do surprisingly well to match the AOD that was derived by the VolMIP models. This gives credibility to the discussions comparing and contrasting global stratospheric average values of sulfate burden and effective radius across models in the results Sect. 3.

### 4.2 Major simplifying assumptions made in models which caused these differences

Next, we look at why the models disagree on sulfate burden and effective radius. For a fixed size distribution, mass burden, and mass fraction of sulfuric acid within the sulfate aerosol ($\omega$), the number of optically active particles should vary by a factor of $\frac{1}{Reff^3}$. For the *Reff* difference between Groups AODHigh and AODLow, this translates to about a factor of 2.5 variation in number. The global aerosol mass is almost the same for the various models six months after the eruption except for LMDZ-S3A (Figure 2, Figure 7b), but the effective radius varies from about 0.7 μm for MAECHAM5-HAM point to about 0.45 μm for CESM-WACCM
and UM-UKCA (Figure 3). It thus follows that either the width of the size distributions is highly variable between models (Sect. 4.3.1), or the number of particles is highly variable. More, smaller particles could be generated by a faster nucleation rate, a more prolonged period of new particle formation (Sect. 4.2.1), or a slower coagulation rate perhaps due to more rapid dispersion of the cloud over the planet (Sect. 4.2.2). Unfortunately, it is difficult to use particle number, which was not a variable output by the models in this experiment, as a parameter to understand optical properties because there can be large numbers of particles in freshly
nucleating clouds that are optically ineffective. A third option is that the models differ in their handling of $\omega$, which is discussed in Sect 4.3.3.

### 4.2.1 Interactive OH

The rate at which SO$_2$ is converted to sulfate, which is controlled by the OH abundance, impacts the particle effective radius in a number of ways. Rapid production of sulfuric acid leads to high nucleation rates and high growth rates, which ultimately lead to



larger particles. Slow production of sulfuric acid reduces the nucleation and growth rates, generally leading to smaller particles. Table 2 shows which models include interactive OH chemistry. After a large volcanic eruption, the reaction of $SO_2$ with OH locally depletes the concentration of OH, which is a limiting reactant in the conversion from $SO_2$ to $H_2SO_4$. These reductions are not small. Zhu et al.'s (2020, accepted) WACCM simulations have a reduction of a factor of 2 in OH in the volcanic plume one day after the small 2014 Mt. Kelut eruption (VEI of 4, stratospheric injection of ~ 0.2 - 0.3 Tg $SO_2$), while Mills et al. (2017, Private

communication) find a >95% reduction in OH in the first weeks of the evolving Pinatubo plume. However, although the chemistry is simple, there are no measurements of the OH depletion in volcanic clouds, and for that matter OH is not directly measured in the lower stratosphere. LeGrande et al. (2016) suggested that volcanic water injections could be important for OH. The reaction of $O^1(D)$ with water frees OH and counteracts the OH-depletion by $SO_2$. By supplementing OH mixing ratios, an injection of water into the stratosphere from an eruption could reduce the impact of limited OH on stratospheric chemistry. However, in modeling

studies of the Toba supervolcano eruption for a $SO_2$ injection roughly 10 times that of Tambora, Bekki et al. (1996) show that an injection of water does not completely counteract the OH-depletion by $SO_2$. Zhu et al. (2020, accepted) find that orders of magnitude greater water injection than observed from Kelut is needed to provide enough OH to counteract the loss from $SO_2$ chemistry. Interactive OH is still needed in models regardless of whether or not an injection of water also occurs. In the VolMIP-Tambora ISA ensemble experiment there is no injection of water to limit the impact of $SO_2$ depleting OH. Instead of comparing

models to observations, we compare model outputs with each other. In the VolMIP-Tambora ISA ensemble experiment, local depletion of OH occurs in all of the models that have interactive OH chemistry: CESM-WACCM, SOCOL-AER, and UM-UKCA (Marshall et al., 2018). EVA is not an interactive aerosol model, and thus does not include full sulfur chemistry and OH chemistry is not applicable. In MAECHAM5-HAM and LMDZ-S3A, the OH is prescribed in background climatological concentrations and is thus not depleted from the eruption. In studies of the Toba eruption, when interactive stratospheric OH chemistry was included

the transition from $SO_2$ to $H_2SO_4$ was delayed, yielding a longer-lasting peak concentration of sulfate. The limited OH resulted in a longer lifetime of the volcanic cloud (Robock et al., 2009; Bekki et al., 1996; Bekki 1995; Pinto et al., 1989). A study based on Mt. Pinatubo by Mills et al. (2017) using CESM1/WACCM supported the idea that if local depletion of OH occurred within the volcanic cloud of $SO_2$, the $e$-folding decay time for $SO_2$ oxidation was significantly prolonged. We infer that the lack of interactive OH in MAECHAM5-HAM and LMDZ-S3A is a significant cause of why the global sulfate peaks at least three months earlier in

them than in any of the other models. In Sect. 4.2.2 we discuss the impacts that an earlier production of sulfate has on $Reff$ in a band vs. point injection.

### 4.2.2 Grid cell ("point") vs. zonal ("band") injections

The inclusion of band injections was performed to determine if the initial spatial distribution of the volcanic injection matters. The degree to which spatial distribution matters depends on whether the oxidation rate for $SO_2$ is longer or shorter than the several

weeks needed for the $SO_2$ to be transported around the Earth and become partially homogenized. If the $SO_2$ oxidation time is short, then the nucleation rate, coagulation rate and growth rate would also need to be fast for there to be a difference between the results of point and band injections. Since we see the sulfate forming soon after the $SO_2$ is lost in the VolMIP models, the nucleation rate, coagulation rate and growth rate are rapid because the bulk of the sulfur is not sitting in the $H_2SO_4$ vapor phase. The difference between point and band injections in SOCOL-AER is insignificant, probably because the $SO_2$ stratospheric lifetime in the

experiment in this model is longer (2 month $e$-folding decay time) than the time needed to transport the $SO_2$. As a result, the gas from a point injection can form a band before much sulfate is produced from the oxidation of $SO_2$. However, in MAECHAM5-HAM the band injection experiment has an AOD twice as high as its point injection, which is ultimately due to the short stratospheric lifetime of the $SO_2$ in this model (<1 month $e$-folding time), which is on the same time scale as the transport time.



For the band injection, the lower concentration of sulfuric acid and water vapor presumably causes less nucleation and
condensational growth than for the point injection and the corresponding lower concentration of the sulfate aerosols leads to less coagulation. As a result, the band injection experiment in MAECHAM5-HAM produces aerosol particles with smaller effective radii, which are more efficient optically and have lower falling velocity, thus resulting in higher AOD with a longer *e*-folding time.

The geoengineering studies by Niemeier et al. (2011) and Niemeier and Timmreck (2015) using ECHAM5-HAM reached the
opposite conclusion; they found that increasing the injection area by using a band injection instead of a point injection resulted in larger *Reff* and lower AOD. These geoengineering studies noted that the lower concentration of $SO_2$ and more equally distributed $H_2SO_4$ in their band injection (interactive OH was not included in their simulations) led to condensation occurring on pre-existing particles rather than to nucleation, causing lower particle numbers with larger *Reff*. However, the geoengineering studies were for continuously emitted $SO_2$ at rates of 4 and 10 TgS of $SO_2$ per year, which is much lower in concentration than the 60 TgS of $SO_2$
injected over 24 hours in this VolMIP-Tambora ISA study, and still lower in concentration than more common Pinatubo-sized volcanic events due to the temporal emission scale. Continuously emitting $SO_2$ instead of injecting $SO_2$ in pulses can significantly affect the size of the sulfate particles (Heckendorn et al., 2009) because they add to the particles already present rather than making many more. Similarly, a volcanic injection into high sulfate background levels would result in larger particles (Laakso et al, 2016). Volcanic eruption studies such as this VolMIP-Tambora experiment inject $SO_2$ into low sulfate background concentrations. Thus,
the use of geoengineering studies as an analog for volcanic eruptions should be taken with care. Generalizations from geoengineering studies in terms of the results of horizontal injection area should not be applied to modeling volcanic events, as we find that volcanic injections of $SO_2$ into low sulfate background concentrations give opposite results between using band injections compared to point injections than do geoengineering studies of continuously injected $SO_2$.

**4.3 Other model uncertainties**

The CESM-WACCM, SOCOL-AER point and UM-UKCA models do have interactive OH chemistry, and do not use band injections. Yet, their results still vary in AOD, sulfate mass, and effective radius. Further explanations are therefore needed to understand these disparities. Table 2 shows a number of additional differences between the models, which relate to the setup of the model's size distribution, to photolysis, and to stratospheric meridional transport, and may contribute to remaining inconsistencies.

**4.3.1 Size distribution scheme**

First, there are differences in the ways in which the models treat the aerosol size distribution (Table 2, Appendix B). Modal models assume a lognormal size distribution, whose mean size is allowed to vary, but whose width is fixed. Sectional models define the size distribution using a fixed set of size bins, usually resolved over a logarithmic grid, and allow the number concentration within each size bin to vary. Modal models suffer from sensitivity to choice in mode width, and sectional models may not resolve the
distributions well by having too few bins. Kokkola et al. (2009) found the differences in results arising from these limitations to be enhanced with larger volcanic injections of $SO_2$. In a separate study, Weisenstein et al. (2007) performed a global 2-D intercomparison of sectional and modal aerosol models by contrasting 20-, 40- and 150-bin sectional models with 3- and 4-mode modal models in simulations for ambient stratospheric sulfate, and for the Pinatubo volcanic cloud. They found significant errors in using modal models unless care was taken in the width of the modes, and that none of the modal models considered compared
well with the sectional model for effective radius. English et al. (2013) explored the variation of lognormal fits to simulated size distribution and found that the widths change with size of eruption, time and location. A new aerosol microphysics model,





SALSA2.0 (Kokkola et al., 2008; Kokkola et al., 2018), was implemented in another study (Kokkola et al., 2018) as an alternative microphysics model to the default modal scheme in ECHAM-HAMMOZ. They found that the sectional model was able to slightly better reproduce the observed time evolution of the global sulfate burden and stratospheric aerosol effective radius compared to

the modal aerosol scheme in their simulations of the 1991 Pinatubo eruption. We suggest, for each model in this VolMIP-Tamborra ISA ensemble that has the option to use a sectional or modal model in its aerosol size distribution scheme, to run this same Tambora experiment using its counterpart, so that the differences in produced *Reff* from choice of aerosol size distribution scheme might be further assessed. At this time, we cannot make any conclusions about whether the use of modal vs. sectional size distribution schemes plays a role in the intermodel disagreement of the VolMIP-ISA models in this experiment.

**4.3.2 Stratospheric meridional transport**

The VolMIP eruption goes into the tropical pipe, which is a region that has little poleward transport in the summer hemisphere. Within the stratosphere aerosols are transported meridionally towards the winter pole, which drains the tropical pipe. Additionally, the stratospheric optical depth maxima move poleward for the same reason that ozone columns are highest poleward, that is, the stratosphere is twice as deep at mid and high latitudes. Aerosols are removed from the high latitudes by tropopause folding. As

transport next occurs towards the other pole during its winter, the tropical pipe is again depleted. The models differ in simulated stratospheric meridional transport of the volcanic aerosol. Figure D1 shows the time evolution of stratospheric meridional circulation patterns of aerosol in terms of AOD. The volcanic aerosol that is injected into the tropical stratosphere is transported poleward by the Brewer-Dobson circulation. SOCOL-AER transports its aerosol to the Southern Hemisphere earlier than the other models, and MAECHAM5-HAM is the first to transport the bulk of its aerosol to southern high latitudes, which is where the polar

vortex is located and tropopause folds provide a sink. The bulk of the aerosols in LMDZ-S3A, CESM-WACCM and UM-UKCA remain in the tropics for the longest time before being transported towards southern high latitudes. After the meridional profile of AOD first reaches a maximum at southern high latitudes, the location of maximum AOD alternates hemispheres with season, peaking at high latitudes. The assumed mechanism for this observation of the model results is that remaining aerosols in the tropics, within the subtropical barriers, follow the seasonal movement of the tropical pipe and are transported poleward as it drains. EVA

is not a model in the GCM sense, and its method for simulating stratospheric aerosol distribution is to separate the stratosphere into three zonal regions – equatorial, Northern Hemisphere extratropical, and Southern Hemisphere extratropical – and describe the stratospheric aerosol distribution as the superposition of three zonally symmetric, global-scale aerosol plumes (Toohey et al., 2016). With the exception of EVA, part of why the models differ in stratospheric meridional circulation patterns in this study may be due to their different approaches in the treatment of the QBO (Table 2), and/or to differences in transport vertical diffusion

associated with the various vertical model resolutions and amount of vertical levels in the volcanic model (Table 1). All of the models except for EVA include aerosol influence on radiation, which warms the aerosol layer, which forces self-lofting and latitudinal spread (Young et al., 1994; Timmreck and Graf, 2006). Meridional transport may also simply be faster or slower depending on the internal model dynamics. For example, outside of this study, ECHAM5, the GCM used by both SOCOL-AER and MAECHAM5-HAM, has been documented to have a too-fast vertical ascent and/or mixing in the lower tropical stratosphere

(Stenke et al., 2013) and too-fast poleward transport in the stratosphere from the tropics (Oman et al., 2006). Also, Niemeier et al. (2020) show that in the lower tropical stratosphere around 50 hPa, WACCM has 70% larger residual vertical velocity than ECHAM5. Simulations with ECHAM5 and WACCM in Niemeier et al. (2020) where the QBO is internally generated show that stronger residual vertical velocity strengths and subsequent vertical advection strengths can lead to different tropical sulfate altitudes, concentrations, and meridional stratospheric transport.




The CESM-WACCM runs provide insight about the relative importance that stratospheric meridional transport speed actually has on the global stratospheric AOD. For the 2 CESM-WACCM runs using the easterly 1982 QBO forcing, the aerosol remains concentrated in the tropics for longer (Figure D2 in red) than for the 3 CESM-WACCM runs that used the easterly 1991 QBO forcing (Figure D2 in blue). In the CESM-WACCM runs using the 1991 QBO forcing, aerosol is transported more quickly from

the source of the eruption in the tropics to the southern extratropics. Due to the complexity of stratospheric dynamics, we do not attempt to draw conclusions here about the degree to which the treatment of the QBO specifically affects the simulated stratospheric meridional transport patterns. Instead, we focus on the different stratospheric meridional transport patterns which are produced and their impact on AOD. The 2 CESM-WACCM runs (labeled in red in Figure D2) that have aerosol remain in the tropics for longer produce larger $Reff$, and a longer global mean AOD $e$-folding time (Figure 8 labeled in red) than the 3 CESM-WACCM runs

(Figure 8 labeled in blue) where aerosol is more quickly transported poleward. Thus, the CESM-WACCM ensemble runs show that there is an influence of meridional circulation on stratospheric global mean AOD. However, the difference between resultant global AOD outputs arising from the two meridional stratospheric circulation patterns found in the CESM-WACCM runs shown in Figure 8 is minor compared to the intermodel disagreement on global AOD displayed in Figure 1. While further investigation on stratospheric circulation is beyond the scope of this paper, we conclude from this analysis that the impact of inter- and intra-

model meridional stratospheric circulation discrepancies alone within this experiment on the global mean stratospheric AOD are small compared to the larger issues that are caused by simplified aerosol chemistry and by disagreement in $Reff$.

### 4.3.3 Aerosol composition

The values of $\omega$ (and thus $\rho$) vary strongly with altitude because they depend on the water vapor concentration. As particles grow larger by coagulation and condensation of sulfuric acid they drift downward and the mass fraction of water in the $H_2O$-$H_2SO_4$

aerosol droplet grows (i.e., $\omega$ decreases). This is most clearly seen in the plots of CESM-WACCM and SOCOL-AER in Figure 9. Up until around September of 1815, the dashed lines of real AOD for CESM-WACCM and SOCOL-AER best match the shading marking the reconstructed AOD from Eqs. (1 and 2) where $\omega = 0.9$ is used, and as time progresses and the particles swell up with water and grow in size, $\omega$ is decreasing until the dashed lines of real AOD match the shading for the reconstructed AOD where $\omega = 0.75$ is used.


In the VolMIP models, the water on the particles is found by assuming the particles are in equilibrium with the water vapor partial pressure. The water vapor mixing ratio is approximately independent of altitude in the lower stratosphere, so the partial pressure decreases exponentially between the tropopause and the 26 km injection height assumed in VolMIP. As a result, the product of $\rho$ and $\omega$ should decrease by a factor of about 2 as the particles move from their injection height to the tropopause and swell up by

picking up additional water (Figure S2).

Variation of $\omega$ has a significant impact on AOD. Although we do not know the actual values of $\omega$ (and $\rho$) calculated in the VolMIP models, we do have information about the ways in which they are calculated. The most prominent difference between VolMIP model physics is that MAECHAM5-HAM does not allow $\omega$ to vary spatially or temporally. Instead, MAECHAM5-HAM assumes

a prescribed $\omega = 0.75$ throughout the entire stratosphere. LMDZ-S3A uses $\omega = 0.75$ for calculation of refractive index for $q$, but otherwise allows $\omega$ to vary spatially and temporally. The sources used for the calculations of $\omega$ and $\rho$ in the VolMIP models are listed in Appendix E.



### 4.4 Missing processes

#### 4.4.1 Inclusion of aerosol scattering in photolysis calculations

We already discussed the role of OH depletion by $SO_2$, and the need to calculate OH interactively. For large $SO_2$ injections, Pinto et al. (1989) showed that $SO_2$ shielding ozone via UV absorption could impede ozone photolysis, thereby impacting OH via an additional mechanism and thus impacting the $SO_2$ oxidation rate. However, none of the models in VolMIP considered this effect. None of the models include the direct reduction in solar radiation from aerosol scattering in their photolysis and photorate schemes either. CESM-WACCM, SOCOL-AER, and MAECHAM5-HAM use lookup tables which depend only on overhead ozone and

molecular oxygen to compute photorates. Since these models ignore the volcanic aerosol, which can be optically thick (as demonstrated by the AOD values reached here), they may significantly err in their calculations of photolysis. UM-UKCA uses Fast-J (Wild et al., 2000) and Fast-JX (Neu et al., 2007; Prather et al., 2012) photolysis schemes, but unfortunately they did not include aerosols in these schemes. Effects of volcanic aerosols on photolysis rates have been looked at before (Timmreck et al., 2003; Rozanov et al., 2002; Pinto et al., 1989), but a detailed estimate of what the impact of volcanic aerosols on photolysis would

be in these simulations is missing. MAECHAM5-HAM prescribes OH, so the effect of aerosols on photolysis is irrelevant here. Photolysis effects are not included in LMDZ-S3A. The importance of this exclusion of aerosols in the photolysis calculations in CESM-WACCM, SOCOL-AER and UM-UKCA on the resultant AOD is yet to be determined, and it is possible that the significance may vary by model and by the optical depth of the particles. The CESM-WACCM group is working on an interactive version of the radiation code to test the importance of the volcanic aerosol to the photorates.

#### 4.4.2 Volcanic ash

Another factor for consideration is that this experiment excludes volcanic ash injections. Fine ash particles have a direct radiative forcing effect. Pueschel et al. (1994) state that the mixed ash/sulfate particles increase the particulate surface area up to 50-fold after the 1991 Pinatubo eruption. Volcanic ash containing particles have been observed 8 months after the 1991 Pinatubo eruption (Pueschel et al., 1994), and one year after the 1963 Mt. Agung eruption (Mossop 1964) and the 1982 El Chichón eruption (Shapiro

et al., 1984). Ash can reduce the available solar radiation for photochemistry. As none of the models even take aerosols into account for impacting photorates, neglecting ash would not directly alter intemodel disagreement on photorates. Buoyancy changes from radiative heating of the dark ash can cause self-lofting of the volcanic cloud. The different altitudes of the volcanic cloud may alter photolysis rates. The VolMIP-Tambora ISA experiment protocol prescribed an injection altitude for the volcanic cloud, which in theory should remove this potential indirect source of intermodel disagreement from excluding ash, by acting as if the self-lofting

had already occured. Due to the process of ash scavenging, injected ash can decrease the $SO_2$ (e.g. Zhu et al., 2020, accepted) and sulfate (Muser et al., 2020) residence time and concentration in the stratosphere. The result would be a lower sulfate aerosol mass burden, and possibly altered size distribution. While ash scavenging would affect model results compared to observed quantities of sulfate burden and AOD, neglecting the injection of ash should not be a direct source of intermodel disagreement in this VolMIP-Tambora ISA ensemble experiment due to the coordinated injection protocol. All models began their runs with the same mass

burden of $SO_2$, which could be thought of as all starting the experiment with the same $SO_2$ after ash scavenging had already occurred.

#### 4.4.3 Consequences of missing processes for the Tambora-ISA ensemble and others

Since the VolMIP-Tambora ISA experiment protocol assigns a coordinated injection altitude and quantity of $SO_2$, the assumption was that the missing processes of ash, volcanic water injections, and aerosols (and ash) impacting photolysis rates should not be





consequential to intermodel disagreement on AOD because none of the models calculate these processes. In practice, however, there is potential for intermodel differences in the indirect consequences because some models fully calculate certain processes, while others use simplifications based on observations. The impacts of ash, water, and aerosol on photolysis rates which are occuring in reality can be ingrained in the observations that some models are basing parameterizations on. For example, the $e$-folding lifetime for $SO_2$ can be reduced by oxidation on ash and by ash scavenging, or increased by the impact of aerosols and ash

on photolysis rates. The composition of the sulfate aerosol ($\omega$ and $\rho$) can be impacted if volcanic water is injected, which alters the ambient relative humidity and thus water content of the aerosol. The consequence then is that effects of these "missing processes" could actually still be included in a heavily parameterized mode such as EVA, while being specifically excluded in the aerosol microphysical models. The degree to which this matters depends on how large of a role the missing processes play in the observations used in the parameterizations, and how closely the observations for those parameterizations would be applicable to

the specific volcanic injection simulated.

The important factor for whether there will be a difference between using a band, area or point injection is whether the sulfate aerosol is being produced before the volcanic cloud has time to spread to the larger area. Guo et al. (2004b) report that half of the sulfate in the 1991 Pinatubo cloud formed in the first four days. Satellite data showed that the fastest $SO_2$ decay rate was occuring

in the first five days after the Pinatubo eruption (Guo et al., 2004b), when $SO_2$ was still very high and hence OH should be low. The rapid decay of $SO_2$ was possibly due to heterogeneous oxidation on volcanic ash, but the volcanic $SO_2$ decay is still not completely understood. The $SO_2$ stratospheric $e$-folding lifetimes produced in this experiment in MAECHAM5-HAM, which uses constant prescribed stratospheric concentrations of the oxidants OH and ozone, are similar to the upper tropospheric/lower stratospheric $e$-folding times observed following the eruptions of Pinatubo and other moderate-sized eruptions (Carn et al., 2016).

For small injections, Carn et al. (2016) show from satellite measurement that the $e$-folding lifetime of $SO_2$ in the upper troposphere/lower stratosphere can be even shorter: on the order of a week. It is difficult to deduce from measurements of $SO_2$ alone what the stratospheric oxidation time is for conversion to sulfate, particularly if the observations are not restricted to above the tropopause. We deduce from the VolMIP-Tambora ISA experiment that for volcanic events which produce short $SO_2$ oxidation times, band injections produce smaller $Reff$ and larger AOD than point injections. This is what we see from the MAECHAM5-

HAM runs. For very large volcanic eruptions like Tambora, if interactive OH is simulated, then band injections might be able to pass as representative of point injections due to the long $SO_2$ oxidation time that is caused from OH-depletion by $SO_2$, which is what we see in the SOCOL-AER runs. What we do not know, however, is a specific cutoff in volcanic injection size that would allow a band or area injection to work well in replacement of a point injection, partly because we would need to know how much the missing processes in this Tambora-ISA ensemble impact the $SO_2$ decay rate.

**5 Conclusions**

We sought to answer the question: why do the VolMIP-Tambora ISA models drastically disagree on global stratospheric AOD under a coordinated injection experiment protocol designed to eliminate confounding variables? We have identified physics and chemistry that some models handled differently, made simplifying assumptions about, or even left out entirely, which contributed to the intermodel disagreement on the $Reff$ and stratospheric sulfate burden, and therefore led to a wide range of simulated

magnitude and duration of the volcanic perturbation to AOD.





*Reff* and sulfate mass are key variables in the AOD equation, Eq. (1). At times when the models agree on the amount of sulfate in the stratosphere, they disagree on the corresponding magnitude of stratospheric AOD because the aerosols have different *Reff* (Table 4, Fig. 5). Thus, particle size is a main source of AOD disagreement during the first year. The rise and decay of sulfate

aerosol burden in the stratosphere controls the timing of the onset and duration of elevated AOD. Differences in the simulated sulfate burden is the factor which is most responsible for intermodel disagreement in AOD after the first year (Fig. 7). However, the *e*-folding time of sulfate burden is influenced by *Reff* because sedimentation depends on particle size.

The values of $\omega$ and $\rho$, which are determined by ambient temperature and water vapor pressure, impact the particle radius because

of the contribution of water within the aerosol. The number of particles is controlled by the balance between nucleation and coagulation. For constant sulfur mass, a varying *Reff* indicates that the number of particles must be different between the model simulations, and/or the mass fractions of water (1 - $\omega$) within the sulfate aerosols are different. If the models are complete (or at least consistent) in their governing aerosol microphysics for computing $\omega$ and $\rho$, these processes of nucleation and coagulation must be being treated differently, or be being affected by factors such as transport differently in the models. Coagulation is affected

if the aerosol is spread over a larger geographical area than a more confined one due to the difference in concentration. Stratospheric sulfur chemistry controls the rate of sulfate aerosol production, which in turn can influence aerosol *Reff* if production occurs when the volcanic cloud is still dense. Neither MAECHAM5-HAM nor LMDZ-S3A used interactive OH in their stratospheric sulfur chemistry schemes. Results from the MAECHAM5-HAM point and band experiments show that effective radii will be larger if the conversion to sulfate occurs quickly before the volcanic cloud has dispersed zonally. The MAECHAM5-HAM and LMDZ-

S3A conversion times to sulfate in this experiment are similar to the conversion times following eruptions of the size of 1991 Mt Pinatubo and 1982 El Chichón, which suggests that 2-D injections should not be used for eruptions of those sizes or smaller. The MAECHAM5-HAM point injection greatly differs on *Reff* and AOD compared to its band injection experiment. Initializing a volcanic sulfur injection as a zonal band of $SO_2$ across the globe is unrealistic, as are area injections over many latitudes as used in several studies e.g. Pinatubo eruptions (Dhomse et al., 2014; Mills et al., 2016; Sukhodolov et al., 2018). However, the experiments

from SOCOL-AER imply that a band (2-D injection) and point (3-D injection incorporating longitude) may yield similar results if the conversion time from $SO_2$ to sulfate is longer than the time it takes for stratospheric transport to zonally homogenize a point injection of $SO_2$.

Models with interactive OH chemistry show a strong initial response to the effect of locally depleted OH within the first few

months, which influences the $SO_2$-to-sulfate conversion rate. Volcanic water vapor emissions can supplement the OH mixing ratio, but Zhu et al. (2020, accepted) show that even small eruptions can require very large injections of water to offset this depletion. There is also potential for the water vapor concentrations to increase due to stratospheric heating following an eruption, which can assist the water injections in supplementing the OH mixing ratio. However, large quantities of water dilute the sulfate aerosol, decreasing the value of $\omega$ and making *Reff* unrealistically large if too much water is injected with the model in attempts to offset

the OH depletion (Zhu et al., 2020 accepted). Eruptions that inject greater amounts of $SO_2$ into the stratosphere should have prolonged conversion times to sulfate, because the OH is locally depleted. With a large enough volcanic cloud, $SO_2$ can zonally circulate around the globe more quickly than it is oxidized. The errors induced from the simplification of using prescribed $SO_2$ conversion times or prescribed OH based on observed conversion times from Pinatubo-sized and smaller eruptions will increase for larger volcanic injections. Conversely, if interactive OH chemistry is included, then for larger eruptions the error caused by the

simplification of using a 2-D band injection may be less substantial. Still, it may not be sufficient to compute photorates without including the volcanic aerosols, an issue that needs further study.


In this VolMIP ISA experiment based on Mt. Tambora we find that prescribing a band injection scenario and/or not computing OH interactively causes large differences in the spatio-temporal evolution of stratospheric volcanic sulfur species and the *Reff* of

sulfate aerosols. MAECHAM5-HAM point, MAECHAM5-HAM band, LMDZ-S3A band, and SOCOL-AER band, all have at least one of these simplifications that impact their simulation of AOD. The LMDZ-S3A model also produces very large particles early in the simulation, which we speculate in Appendix C is due to the use of very different nucleation rate expressions from those used in other models. Referring back to Figure 1-3, CESM-WACCM and UM-UKCA have similar *Reff* values and similar masses until 1816, after which CESM-WACCM has larger particles and more mass. However, CESM-WACCM has larger optical depths

even in 1816 and afterward. We have not been able to identify the sources of the differences between UM-UKCA and CESM-WACCM. Nor have we been able to identify why the *Reff* in SOCOL-AER is larger (and thus AOD much lower) than in CESM-WACCM and UM-UKCA. These differences may be due to the much higher model top in CESM-WACCM, to the much higher vertical resolution but slightly lower horizontal resolution in UM-UKCA compared to CESM-WACCM and the much lower horizontal resolution in SOCOL-AER, to differences in model dynamics, to differences in the computed $\omega$ and $\rho$, or to other factors

we have not explored.

This VolMIP-Tambora ISA ensemble exercise on a coordinated large equatorial stratospheric injection of volcanic $SO_2$ has revealed existing deficiencies in advanced models that are highly influential to simulations of aerosol optical depth from volcanic eruptions. Furthermore, it provides insight into the circumstances in which the magnitude of the $SO_2$ burden and dispersion rate of

the advected volcanic cloud would have the greatest impact on the resultant AOD calculations. The different nudged QBO runs by CESM-WACCM show that there is an impact of differences in meridional transport to resultant global stratospheric AOD, but suggest that alone it is small in comparison to other issues in other models. Nudged meteorology could be used in future studies to lessen the impact from differences in meridional transport processes between models to further isolate differences in aerosol evolution, but would come with other caveats. We suggest further work that could be done to resolve the importance of some of

these different parameterizations through more model diagnostics and intercomparison studies, such as the proposed experiments in ISA-MIP (Timmreck et al., 2018), including using a passive tracer to better distinguish between microphysical and chemical effects vs. transport issues. To further test if band and/or area injections will work for large eruptions, full 3-D models such as CESM-WACCM, UM-UKCA, and SOCOL-AER that have long conversion times of $SO_2$ to sulfate in their experiments would need to provide band and area injection experiment runs for comparison. Band and point injections in this VolMIP-Tambora ISA

ensemble introduce differences caused in the microphysics and chemistry of a high concentration vs low concentration volcanic cloud, but the story does not end with "point injections". A point injection in this experiment is really a grid cell injection, and there is subgridscale physics occurring in the plume which is being ignored, but may have an impact on the microphysics. Another general problem is the grid size dependence of processes that are already included in the models, so a high resolution run of the same injected mass of $SO_2$ could give different results. We do not know how much of the intermodel disagreement on AOD plotted

in Figure 1 is due to the different model grid resolutions (Table 1). There are still uncertainties that need to be resolved regarding the importance of including the volcanic cloud in photorate calculations, and the use of modal vs. sectional models and their size distribution resolutions. Additionally, while not included in any of the models considered, it may be necessary for realistic volcanic forcing estimation to include injections of water and ash to properly model the initial phase of volcanic cloud evolution. Injected water can impact both the size distribution of the sulfate aerosol through microphysics, and OH chemistry. Injected ash could also

impact photolysis as discussed in Sect 4.4. We do not yet know how sensitive the models are to injections of water and ash. Thus, it would be interesting to see if including ash and water injections would alter the magnitude of intermodel disagreement on AOD.





Their inclusion, along with new photolysis schemes influenced by the volcanic cloud, would allow for models to reasonably be compared to observations, which would ultimately be the best gauge of model performance.

**Data**

This report is based on the monthly mean data from model outputs uploaded to an external server hosted at LOCEAN/IPSL for use of the VolMIP study. Output from the model simulations used for the present study are accessible from this server upon demand. The majority of post processing in this study was done using Python2.7 and PyNGL *http://www.pyngl.ucar.edu/newusers.shtml*. The machine learning technique of Self-Organizing Maps is used for analyzing the stratospheric meridional circulation patterns shown in Figures D1 and D2 of the supplementary material, which are plotted in

Matlab2018 using the package SOM Toolbox2.0 http://www.cis.hut.fi/somtoolbox/.

**Appendix A: Post processing**

The participating models in this study provided monthly outputs of time averaged data in different vertically and horizontally resolved grids, units and formats. Some values were already pre-processed, for example provided in terms of stratospheric values and/or zonal means. Some gave data for all model levels including the troposphere, and at all longitudes. Some models output

pressure levels as vertical indices, some gave only altitude, and a few provided both. For consistency, the following post-processing methods were applied to obtain the monthly stratospheric global mean values of AOD, sulfur species burdens, and effective radius.

Stratospheric AOD is calculated by integrating extinction with pressure from the top of the atmosphere to the tropopause at the specified visible wavelength in Table 1. The vertical integral is calculated via dot product of the extinction at each layer with its

vertical layer thickness, $h$, given by the hypsometric equation, Eq. (A1):

$$h = \frac{RT}{g} ln \left(\frac{P_2}{P_1}\right)$$                    (A1)

where $R$ is the dry air gas constant 287 J Kg$^{-1}$ K$^{-1}$, $g$ is acceleration due to gravity at sea level 9.81 m s$^{-2}$, and $T$ is the average temperature of the layer (between pressure levels $P_2 < P_1$).

Monthly stratospheric burdens of sulfur species (SO$_2$, H$_2$SO$_4$, SO$_4$) are provided by all models. Effective radius of the wet spherical aerosols ($r_{eff}$) was output with dimensions of time, vertical level, latitude, and longitude in each model. In these models, effective radius is defined to be proportional to the average volume of the particles divided by the average cross-sectional area.

$$r_{eff} = \frac{\int r \, \pi \, r^2 \, n(r) dr}{\int \pi \, r \, n(r) dr}$$                    (A2)

The global stratospheric mean effective radius (*Reff*) is calculated for each month in Eq. (A3) by the sum of each vertical model

level's global mean effective radius ($Reff_\tau$) weighted with the global level mean density of aerosol surface area and the global level mean thickness. This weighting is done so that the stratospheric mean effective radius calculation is performed over the domain where aerosol is present.

$$Reff = \frac{\sum_{\tau=\tau_1}(SAD*h*Reff)_\tau}{\sum_{\tau=\tau_1}(SAD*h)_\tau}$$                    (A3)



where *SAD* is surface aerosol density (μm$^2$ cm$^{-3}$), and *h* is again the thickness in meters calculated via the hypsometric equation,
Eq. (A1), of the model atmosphere layer between level interfaces. The vertical model level index is denoted by the subscript *τ*,
where *τ$_l$* is the index of the model level nearest to the tropopause.

In order to integrate horizontally, one needs to take account of the varying area of the grid cells. Given the number of latitude
points per hemisphere, *nlat/2*, the function Ngl.gaus computes *nlat*-by-one arrays of the Gaussian latitudes and their weights
http://www.pyngl.ucar.edu/Functions/Ngl.gaus.shtml. For consistency, the various global grids were approached in post
processing using these Gaussian latitudes, *glats*, and their weights divided by two, *gwgts,* so that the sum of the array *gwgts* =1.
This substitution was possible because the values of *glats* were virtually identical to the latitude values of the corresponding model
grids. The latitude dimensions from the models were *nlat* = 192 (CESM-WACCM), 145 (UM-UKCA), 98 (EVA), 96(LMDZ-
S3A), and 64 (SOCOL-AER and MAECHAM5-HAM).

Global means could then be calculated by averaging the zonal values while weighting values in their latitude dimension by *gwgts*.
For the global burden calculations of stratospheric sulfur when data were given in units of mass per horizontal grid area, *gwgts*
were also used: burdens were summed at all longitudes into zonal burdens and then multiplied by gwgts, summed globally, and
multiplied by the surface area of the Earth. Ensemble means are taken from the five runs for each model. The exception is LMDZ-
S3A, which only uses one run.

**Appendix B: Size distribution schemes**

In this study, CESM-WACCM uses the Modal Aerosol Model version with three log-normal modes (MAM3), composed of internal
mixtures (referred to as "mixed/soluble") of soluble and insoluble components in the Aitken, accumulation, and coarse modes (Liu
et al., 2012). Table 2 presents more information on the size distributions used. The modal models used here by UM-UKCA
(GLOMAP-mode aerosol scheme) and MAECHAM (HAM), use geometric log-normal mode size distributions of mixed/soluble
species in the nucleation, Aitken, and accumulation modes for volcanic stratospheric aerosols in these simulations (Mann et al.,
2010; Niemeier et al., 2009; Stier et al., 2005). UM-UKCA uses a 4$^{th}$ log-normal mode that is for accumulation with only insoluble
compounds. This accumulation insoluble-only mode does not have any size limit, and represents meteoric-sulfuric particles ranging
from a few nm (smoke cores) up to a few tenths of a micrometer (sulfate particles with smoke inclusions). CESM-WACCM, UM-
UKCA, and MAECHAM5-HAM all have prescribed mode size distributions defined by fixed mode edge radii (size range) and
mode standard deviations (mode width). Mode number concentrations are re-adjusted as needed so that mode radius remains within
its fixed bounds. UM-UKCA and MAECHAM5-HAM both use the same mixed/soluble mode size distributions. The exception is
that the accumulation mixed/soluble mode in UM-UKCA has a width of 1.40 instead of 1.2. SOCOL-AER and LMDZ-S3A use
sectional models with 40 and 36 size bins respectively. Neighboring size bins differ by volume doubling for the sectional models
used by SOCOL-AER and LMDZ-S3A, meaning that the radius of bin *i* equals to 2$^{1/3}$ times the radius of bin *i* – 1.
The bin radii in SOCOL-AER range from 0.39 nm to 3.2 μm, and range in dry radius from 1 nm to 3.3 μm in LMDZ-S3A. EVA
uses a single log-normal size distribution mode with a standard deviation of 1.20. In EVA the effective radius is proportional to
the one-third power of sulfate mass burden, using a fixed scaling factor chosen to produce the best agreement in terms of the peak
global mean effective radius reached after Pinatubo. Unlike the rest of the models in this study, EVA reports the same effective
radius of volcanic aerosol at all vertical levels of the volcanic cloud.


**Appendix C: Weak nucleation in LMDZ-S3A**

LMDZ-S3A is producing large particles much earlier in the simulation than the other models. We speculate that the nucleation rate may be very low in this model compared to others due to the nucleation rate equation used. Nucleation rates in LMDZ-S3A are calculated with a rate proportional to the square of the sulfuric acid concentration under conditions of large sulfuric acid vapor

concentrations (Kleinschmitt et al., 2017). At conditions of lower sulfuric acid vapor concentrations, nucleation rates in LMDZ-S3A are calculated using Vehkamäki et al. (2002), which at low relative humidity gives nucleation rates increasing with squared relative humidity, and at higher relative humidity gives nucleation rates exponentially increasing with relative humidity. The switch between the squared to the exponential dependency of nucleation rate on sulfuric acid concentration is determined by the size of the critical cluster in LMDZ-S3A. We think the data for nucleation proportional to the square of the sulfuric acid abundance are

from near the Earth's surface, and are due to organics and ammonia stabilizing molecular clusters. Data on nucleation rates in the mid-troposphere do not show this square dependence, presumably because there is not enough ammonia, or organics. In a volcanic cloud we would not expect the square dependence, which would lead to very low nucleation rates compared with exponential nucleation rates. In short, the choice of nucleation rate equations used by LMDZ-S3A, if not switching over to the exponential limit, contributes to slower stratospheric LMDZ-S3A nucleation rates than should be expected in reality, or at least as used by

other models.

If, in the Tambora case LMDZ-S3A has very slow nucleation rates, then the logic which follows is that it is not nucleating many particles and thus particles are growing large in size and rapidly falling out. Thus, we propose that the larger sulfate aerosol particle sizes we are seeing in LMDZ-S3A compared to the other models, is due in part to its handling of the nucleation rates. We suggest

that updating the nucleation code within LMDZ-S3A to make sure that an appropriate (i.e. exponentially increasing) nucleation rate equation is being used in the stratosphere could help solve the problems of the large sulfate aerosol effective radii produced in LMDZ-S3A and the rapid removal of sulfate from the stratosphere in future experiments with that model.

**Appendix D: Figures of meridional stratospheric AOD patterns**

The figures using the machine learning method of Self Organizing Maps (SOM) to analyze the meridional stratospheric AOD

patterns (Fig. D1 and Fig. D2) introduced in Sect. 4.3.2 are shown here.

The patterns chosen in Fig. D1a are determined from SOM trained on the set of all meridional AOD profiles of all months of the ensemble mean from each model out of CESM-WACCM, UM-UKCA, MAECHAM5-HAM point and SOCOL-AER point. EVA patterns are excluded from the training dataset for the SOM algorithm because EVA uses a simplified circulation scheme, and

LMDZ-S3A patterns are excluded from the training dataset because their runs were added later. SOCOL-AER band patterns are excluded from the plot for legibility purposes because they are the same as the patterns of the SOCOL-AER point injection. An example of how to read Fig. D1: at the start of 1815, the AOD profile for UM-UKCA best matches the bottom panel (Fig. D1a). When the eruption occurs in April 1815, the AOD in UM-UKCA is concentrated at the tropics, best matching the top panel in Fig. D1a. Following the purple line forward with time in Fig. D1b, UM-UKCA best matches the second-from-top panel of Fig. D1a

during the second half of 1815. The bulk of the AOD is continuing to shift to southern high latitudes, and by January 1816 UM-UKCA's AOD profile best matches the middle panel of Fig. D1a. From 1816 until the end of the time series, UK-UKCA's AOD oscillates meridionally, while best matching the middle panel and fourth-from-top panel of Fig. D1a.





**Appendix E: Calculation of $\omega$ and $\rho$**

The mass fraction of sulfuric acid within the $H_2O$-$H_2SO_4$ aerosol droplet, $\omega$, is calculated as a function of temperature and ambient
water vapor pressure following Eq. (2) and (3) from Tabazadeh et al., (1997) (CESM-WACCM and SOCOL-AER), and using
Steele and Hamill et al., (1981) (LMDZ-S3A), and Carslaw et al., (1995) (UM-UKCA). In MAECHAM5-HAM, $\omega$ is prescribed
in the stratosphere as $\omega = 0.75$.

The volume density of the sulfate aerosol particle ($H_2O$-$H_2SO_4$), $\rho$, is calculated as functions of $\omega$ and temperature. UM-UKCA
uses Martin et al. (2000), SOCOL-AER uses Vehkamäki et al. (2002), MAECHAM5-HAM uses Vignati et al., (2004), and LMDZ-
S3A uses Kleinschmitt et al. (2017). In CESM-WACCM, $\rho$ from $\omega$ and temperature is calculated from linear extrapolation of the
International Critical Tables (NRC, 1928), which have data between 0 and 100°C. Beyer et al. (1996) confirmed that this data may
be extrapolated linearly to stratospheric temperatures with high accuracy. Results for $\rho$ from the method used in CESM-WACCM
do not appear to be significantly different than if using Myhre et al. (2003), which also extrapolates from the ICT data, but uses a
polynomial function instead of a linear function. The comparisons of the polynomial expressions to that of Myhre et al. (2003) are
shown in Figure S4. Polynomial fit equations used by Myhre et al., 2003 (this study) for obtaining $\rho$ from $\omega$ and temperature have
unimportant differences in values between the models. For example, the largest difference in $\rho$ values for a given $\omega$ in the range
of $\omega = 0.5$ to 0.9 and T= 215 - 245K is $\rho = 1.90$ from SOCOL-AER at (T = 215K, $\omega = 0.9$) vs. $\rho = 1.84$ from LMDZ-S3A at (T =
245K, $\omega = 0.9$). Plugged into Eqs. (1 and 2), using $\rho = 1.84$ vs. $\rho = 1.90$ only gives a difference in a factor of 1.03 for AOD. In
MAECHAM5-HAM, $\rho$, is calculated according to equation 7 of Vignati et al., (2004), which is a function of $\omega$, relative humidity,
and number of sulfate molecules in a particle of average mass for that size distribution mode. We do not have the additional data
available to determine whether the values of $\rho$ in MAECHAM5-HAM are significantly different to the what they would be if
calculated using Myhre et al., (2003) or by the methods used by the other models. The different method used by MAECHAM5-
HAM to calculate $\rho$ may be why the reconstructed AOD using Eqs. (1 and 2) at $\omega = 0.75$ is lower than the real AOD, but we can
only conjecture at this time.

**Author contribution**

MK, CT and DZ initiated the Tambora inter comparison pre-study and designed the experimental protocol. Formal analysis and
data curation were conducted by MC using pre-analysis conducted by SB and VP. Funding acquisition for the work done at the
University of Colorado was done by YZ. MM, WB, SB, LM, VP, TS, MT, FT and NL conducted the model simulations that
performed the experiment. Development and design of components of the model methodology was done by MM (CESM-
WACCM), SD and LM (UM-UKCA), ASt, ER and FT (SOCOL-AER) and MT (EVA). Project administration was led by MK.
MK set up a server to collect and distribute the model outputs. MC prepared and created the data visualizations of the published
work. MC wrote the original draft of the paper and the revisions. Critical review, commentary and editing to the written work were
done by MC, OT, JF, MM, AR, ASc, ASt, CT, DZ, ER, FT, GM, LM, MK, MT, SB, TS, UN, and WB.

**Special issue statement**

This article is part of the special issue "The Model Intercomparison Project on the climatic response to Volcanic forcing (VolMIP)
(ESD/GMD/ACP/CP inter-journal SI)". It is not associated with a conference.



**Acknowledgments**

The work at the University of Colorado by Margot Clyne, Owen Brian Toon, and Yunqian Zhu was supported by National Science

Foundation (NSF) grant PLR1643701. NCAR is sponsored by the National Science Foundation. Computing resources (doi:10.5065/D6RX99HX) were provided by the Climate Simulation Laboratory at NCAR's Computational and Information Systems Laboratory, sponsored by the National Science Foundation and other agencies. Margot Clyne would like to thank Mark Jellinek for mentorship at the University of British Columbia, and thanks the rest of the MJCJ group for discussions benefiting this work. Alan Robock is supported by NSF grant AGS-1430051. Fiona Tummon was supported by the Swiss National Science

Foundation grant 20F121_138017. Matthew Toohey acknowledges support by the Deutsche Forschungsgemeinschaft (DFG) in the framework of the priority programme "Antarctic Research with comparative investigations in Arctic ice areas" through grant TO 967/1-1. Ulrike Niemeier and Claudia Timmreck acknowledge support from the European Union FP7 project "STRATOCLIM" (FP7-ENV.2013.6.1-2; project 603557) and the German Research Foundation (DFG) through the research unit VolImpact (FOR2820): projects VolARC and VolClim. Eugene Rozanov and Timofei Sukhodolov's work was supported by Swiss

National Science Foundation under grants 200021_169241 (VEC) and 200020_182239 (POLE). Slimane Bekki acknowledges the continuous support of Centre National d'Etudes Spatiales (CNES) for the project within the framework of the SOLSPEC programme. MAECHAM5-HAM simulations were performed at the German climate Computer Centre (DKRZ). Sandip Dhomse, Graham Mann and Anja Schmidt received funding via the NERC highlight topic consortium project SMURPHS ("Securing Multidisciplinary UndeRstanding and Prediction of Hiatus and Surge periods"), NERC grant NE/N006038/1. We also

acknowledge funding from the U.K. National Centre for Atmospheric Science (NCAS) for Graham Mann via the NERC multi-centre Long-Term Science programme on the North Atlantic climate system (ACSIS) and the Copernicus Atmospheric Monitoring Service (CAMS), one of 6 services that together form Copernicus, the EU's Earth observation programme. The UM-UKCA simulations were performed on the UK ARCHER national supercomputing service with data analysis and storage within the UK collaborative JASMIN data facility. We acknowledge Mohit Dalvi and Nicolas Bellouin for their involvement with the UM-UKCA

aerosol code.

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


**Tables**

| Table 1: Model Overview | | | | | | |
|---|---|---|---|---|---|---|
| Model | Type | Horizontal resolution: lat x lon | Model top, (# levels ) | Injection region | Optical Depth $\lambda$ (nm) | Reference |
| CESM-WACCM | CCM | 0.95°×1.25° | $4.5\times10^{-6}$ hPa, (70) | point | 550 | Mills et al. (2016) |
| UM- UKCA | CCM | 1.25°×1.875° | 0.004 hPa* (85) | point | 550 | Dhomse et al. (2014) |
| SOCOL-AER | CCM | 2.8°×2.8° | 0.01 hPa, (39) | point, band | 440-690 | Sheng et al. (2015) |
| MAECHAM5-HAM | AGCM | 2.8°×2.8° | 0.01 hPa, (39) | point, band | 550 | Niemeier et al. (2009) |
| LMDZ-S3A | CCM | 1.89°×3.75° | 0.0148 hPa, (79) | band | 550 | Kleinschmitt et al. (2017) |
| EVA | 2-D scaling based idealized volcanic forcing model** | | | | 550 | Toohey et al. ( 2016) |

*\* 85 km. Converted in this table to pressure using 1976 US Standard Atmosphere*
*\*\* EVA output used here is at 1.8° latitude resolution with 31 altitude-defined vertical levels*




**Table 2: Physics and Chemistry Differences of the Interactive Aerosol Models**

| Model | Interactive OH | Aerosol Size Dist. | Photorates Include Aerosols | QBO |
|---|---|---|---|---|
| CESM-WACCM | Yes | modal, 3 modes[c] | No[i] | Nudged |
| UM- UKCA | Yes | modal, 4 modes[d] | No[j] | Internally generated |
| SOCOL-AER | Yes | sectional, 40 size bins[e,f] | No[k] | Nudged |
| MAECHAM5-HAM | No[a] | modal, 3 modes[g] | No[l] | None |
| LMDZ-S3A | No[b] | sectional, 36 size bins[h,f] | No[m] | Nudged |

[a] Climatological concentrations of background OH values have been taken from Timmreck et al. (2003). In the stratosphere, OH, $NO_2$, and $O_3$ concentrations are prescribed from a climatology of the chemistry climate model MESSy (Jöckel et al., 2005).

[b] OH chemistry is not included in the model. In the stratosphere, OH concentrations are prescribed from a climatology of a 2-D stratospheric chemistry climate model (Bekki et al., 1993), giving a stratospheric mean lifetime of about 36 days for $SO_2$.

[c] CESM-WACCM modes {name, radius limits (nm), standard deviation}: {Aitken, (4.35, 26), 1.6}; {Accumulation, (26.75, 240), 1.6}; {Coarse, (200, 20000), 1.2}. Modes are composed of internal mixtures of soluble and insoluble components ("mixed/soluble").

[d] UM-UKCA modes {name, radius limits (nm), standard deviation}: {Nucleation, ( ,5), 1.59}; {Aitken, (5, 50), 1.59}; {Accumulation, (50 ,500), 1.4 }; {Accumulation insoluble, ( -,-), 1.59 }. For volcanic stratospheric aerosols; only mixed/soluble modes are used except for the accumulation-insoluble mode. See Appendix B.

[e] from 0.39 nm to 3.2 μm.

[f] Neighboring size bins differ by volume doubling, meaning that the radius of bin $i$ equals to $2^{1/3}$ times the radius of bin $i - 1$.

[g] MAECHAM5-HAM modes {name, radius limits (nm), standard deviation}: {Nucleation: ( ,5), 1.59}; {Aitken, (5, 50), 1.59}; {Accumulation, (50 ,500), 1.2 }. For volcanic stratospheric aerosols; only mixed/soluble modes are used.

[h] With a dry radius ranging from 1 nm to 3.3 μm (for particles at 293 K consisting of 100 % $H_2SO_4$) (Bekki et al., 1991).

[i] CESM-WACCM uses lookup table for $H_2SO_4$ photolysis by visible light from Feierabend et al. (2006), and $H_2SO_4$ photolysis by Lyman α from Lane and Kjaergaard, (2008).

[j] UM-UKCA uses Fast-JX photolysis scheme by Wild et al., (2000), Neu et al., (2007), Prather et al., (2012), but does not enact the effects of volcanic aerosol on the FAST-JX photolysis rate calculations.

[k] SOCOL-AER uses lookup table for $H_2SO_4$ photolysis by visible light from Vaida et al., (2003) with corrections from Miller et al., (2007) and $H_2SO_4$ photolysis by Lyman α from Lane and Kjaergaard, (2008).

[l] Photolysis rates of OCS, $SO_2$, $SO_3$, and $O_3$ are prescribed based on zonal and monthly mean data sets from a climatology of the chemistry climate model MESSy (Jöckel et al., 2005).

[m] LMDZ-S3A does not include photolysis in its stratospheric chemistry (Kleinschmitt et al., 2017).





**Table 3:** Maximum Values of Global Stratospheric Burdens of Sulfur Species (TgS) and AOD: Max value, Month of injection experiment at which it peaked, *e*-folding time in months from peak value

|  | SO$_2$ | | | SO$_4$ | | | AOD | | |
|---|---|---|---|---|---|---|---|---|---|
|  | Max[a] | Month[b] | *e*-fold[c] | Max | Month | *e*-fold | Max | Month | *e*-fold |
| CESM-WACCM | 25.74 | 1 | 2 | 28.87 | 12 | 16 | 0.67 | 13 | 17 |
| UM-UKCA | 26.99 | 1 | 2 | 26.90 | 7 | 14 | 0.53 | 5 | 17 |
| SOCOL-AER band | 25.24 | 1 | 2 | 27.30 | 8 | 10 | 0.36 | 9 | 11 |
| SOCOL-AER point | 25.27 | 1 | 2 | 26.94 | 8 | 10 | 0.37 | 9 | 11 |
| MAECHAM5-HAM band | 19.55 | 1 | <1 | 28.11 | 5 | 10 | 0.61 | 4 | 11 |
| MAECHAM5-HAM point | 19.36 | 1 | <1 | 28.05 | 5 | 8 | 0.36 | 5 | 10 |
| LMDZ-S3A band | 20.89 | 1 | 1 | 23.03 | 4 | 8-9 | 0.27 | 4 | 10 |
| EVA | NA | - | - | NA | - | - | 0.35 | 9 | 15 |

[a] 30 TgS of SO$_2$ was injected, but data outputs are monthly, so some SO$_2$ has already been removed or converted by the time of the April 1815 data output.
[b] Month index when max value occurs. (Example: April 1815 would be month #1, July 1815 is month #4).
[c] SO$_2$ *e*-folding time is taken in months from a peak value of 30 TgS.


**Table 4:** Global Stratospheric Mean When SO$_4$ > 25 TgS

|  | Effective Radius (μm) | AOD |
|---|---|---|
| CESM-WACCM | 0.47 | 0.63 |
| UM-UKCA | 0.49 | 0.50 |
| MAECHAM5-HAM band | 0.55 | 0.58 |
| SOCOL-AER point | 0.62 | 0.36 |
| SOCOL-AER band | 0.63 | 0.36 |
| MAECHAM5-HAM point | 0.73 | 0.34 |






**Figures**

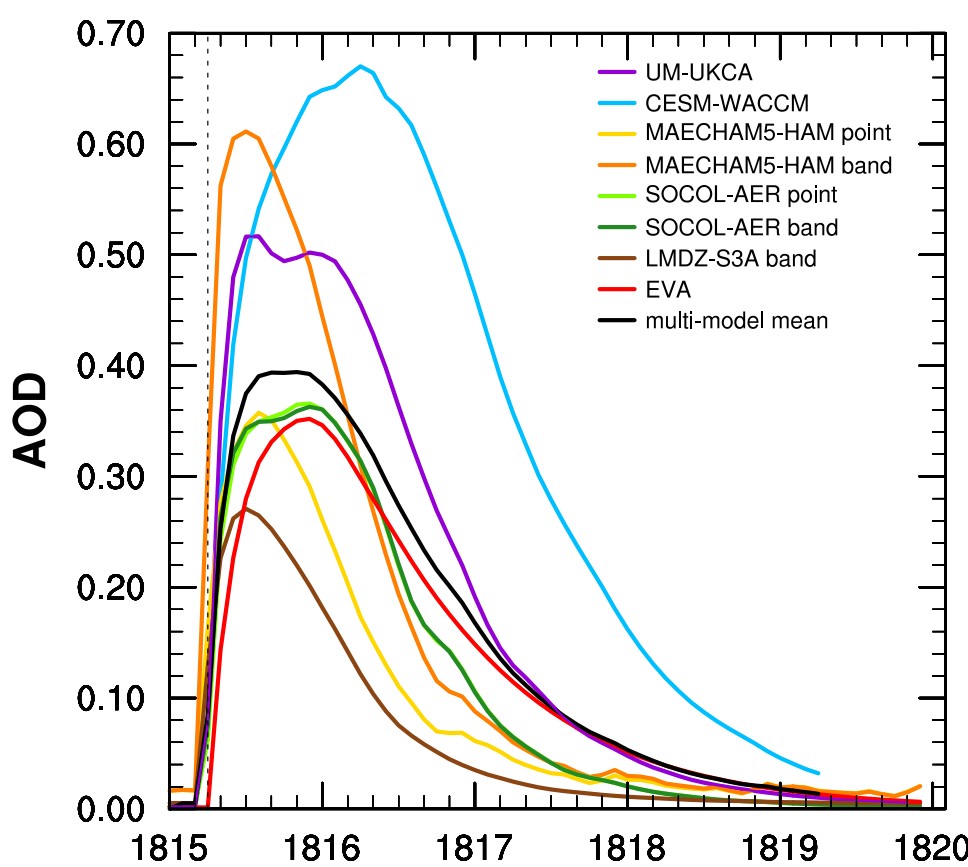

Figure 1: Ensemble mean global mean stratospheric AOD in the visible of participating models. The black line (the VolMIP-Tambora ISA ensemble mean) is the mean of CESM-WACCM (blue), UM-UKCA (purple), SOCOL-AER point (green), MAECHAM5-HAM point (gold), LMDZ-S3A band (dark brown) and EVA (red) models. SOCOL-AER and MAECHAM5-HAM band injection experiments are in green and orange respectively. Vertical dotted line marks date of injection of $SO_2$, which is slightly offset from the zero AOD in the models due to the temporal resolution of the model output and curve smoothing in the plotting program.






**Figure 2:** Global stratospheric burden of SO$_4$ in TgS vs time. Vertical dashed black line indicates month of injection.






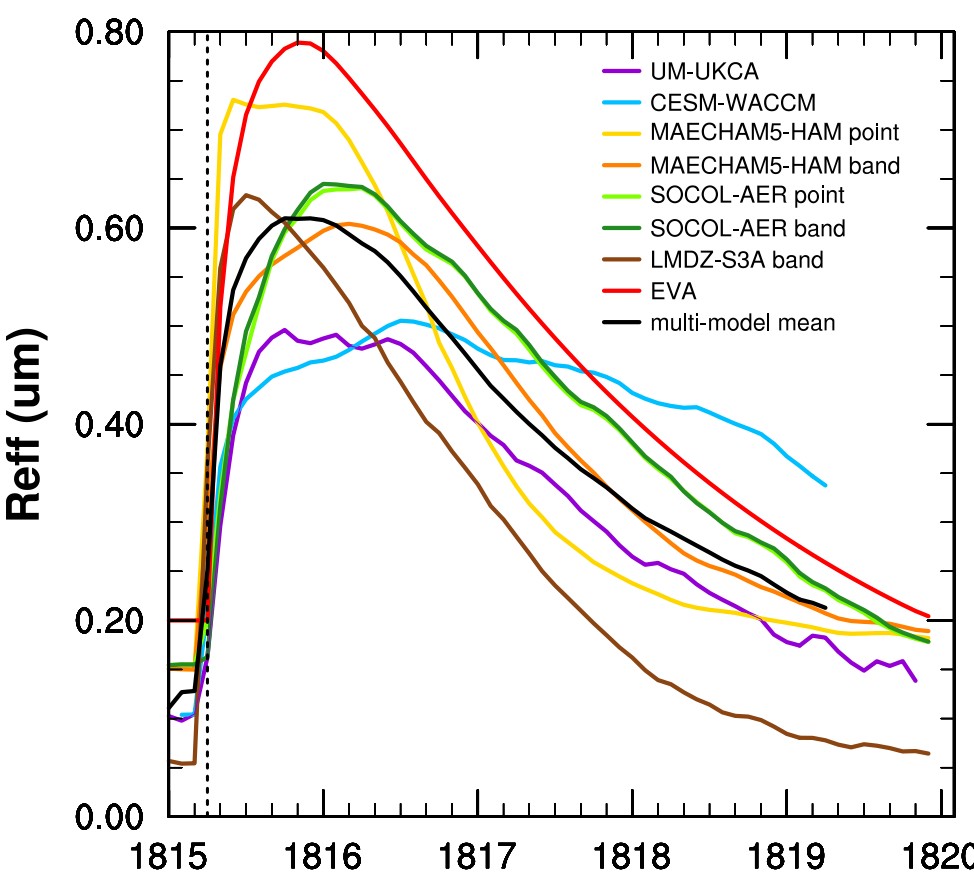

**Figure 3: Global stratospheric mean effective radius (*Reff*) time series. Vertical dotted line marks date of injection of SO$_2$. The calculation of *Reff* is weighted by surface aerosol density and gridcell volume, as explained in Appendix A.**



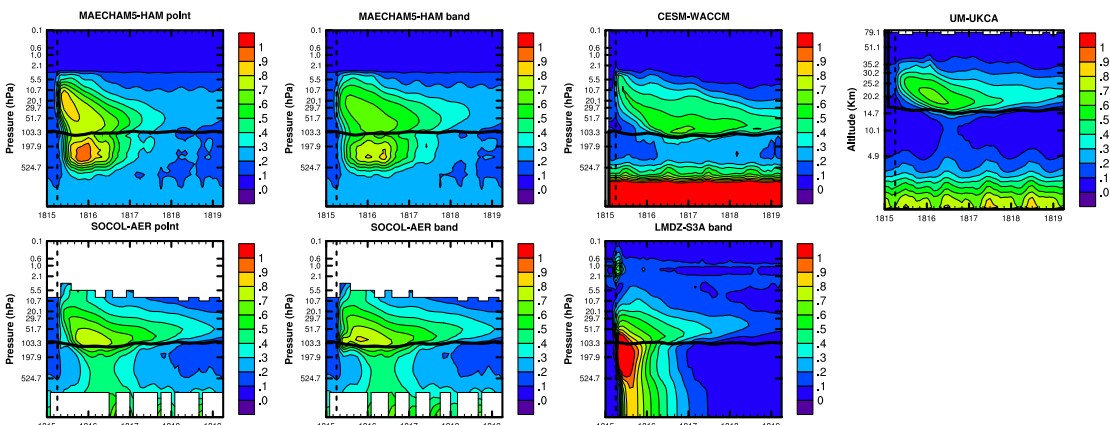

**Figure 4: Vertical profile of tropical mean [23S, 23N] effective radius contours in units of μm marked by the colorbar. Vertical dashed line marks April 1815 injection. Horizontal solid black line marks tropopause height. The large particles in the lower troposphere in Figure 4 (CESM-WACCM and UM-UKCA) are due to background particles such as sea spray and dust.**


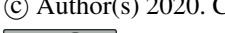




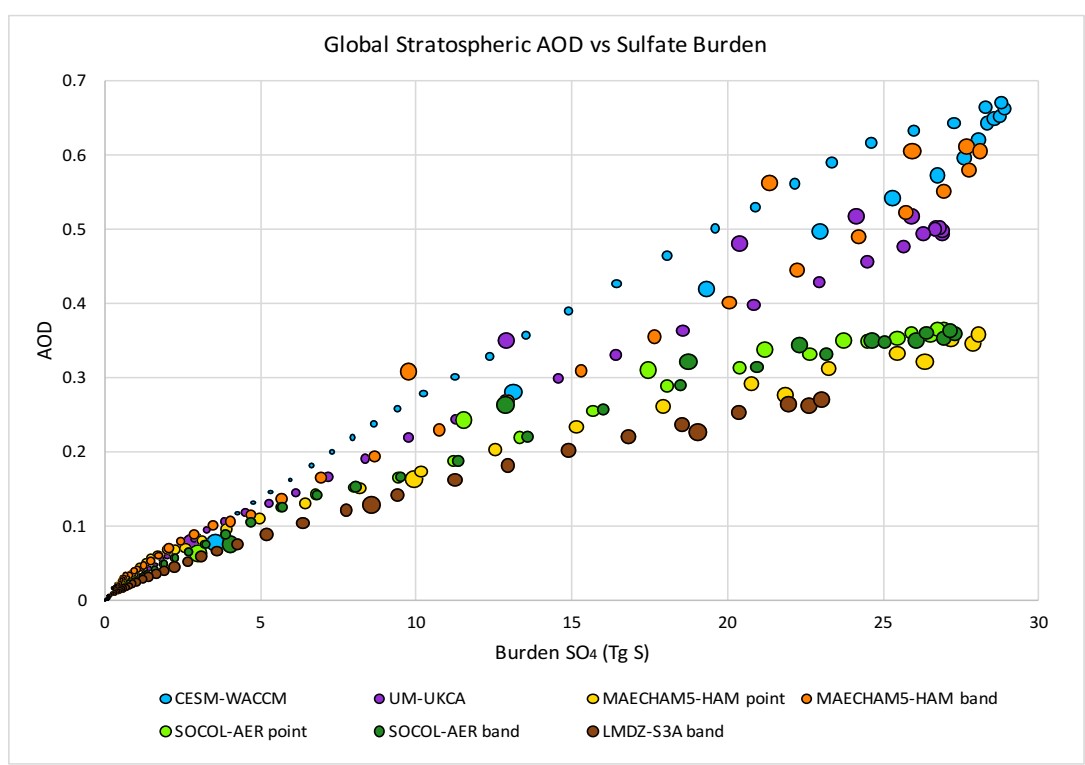

**Figure 5: Global stratospheric AOD in the visible vs sulfate burden from VolMIP-Tambora ISA ensemble means. Circles indicate monthly outputs, and circle marker sizes decrease with age after SO$_2$ injection. Larger circles indicate values occurring from months soon after the eruption date.**






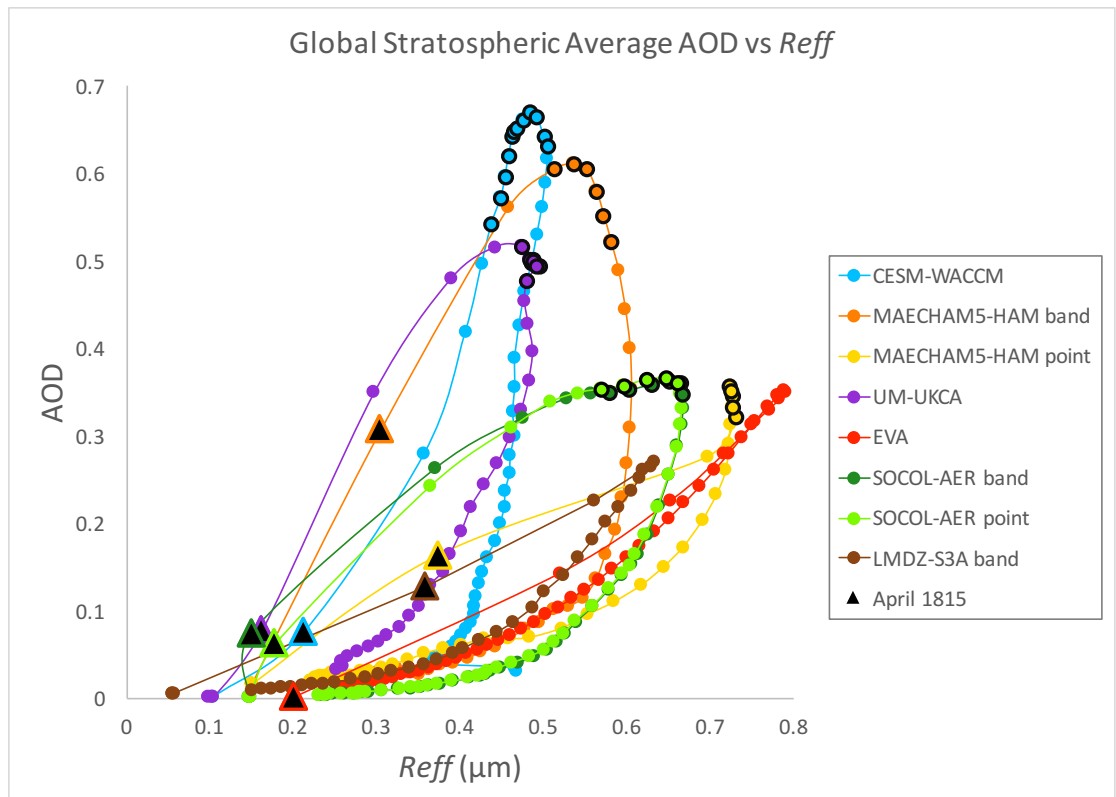

**Figure 6:** **Global stratospheric mean AOD in the visible vs effective radius (µm). Points are connected in order (clockwise) of monthly values from January 1815- April 1819. Circles with black outlines are for months when global stratospheric sulfate burden > 25 TgS. The injection date of April 1815 is indicated by triangles**





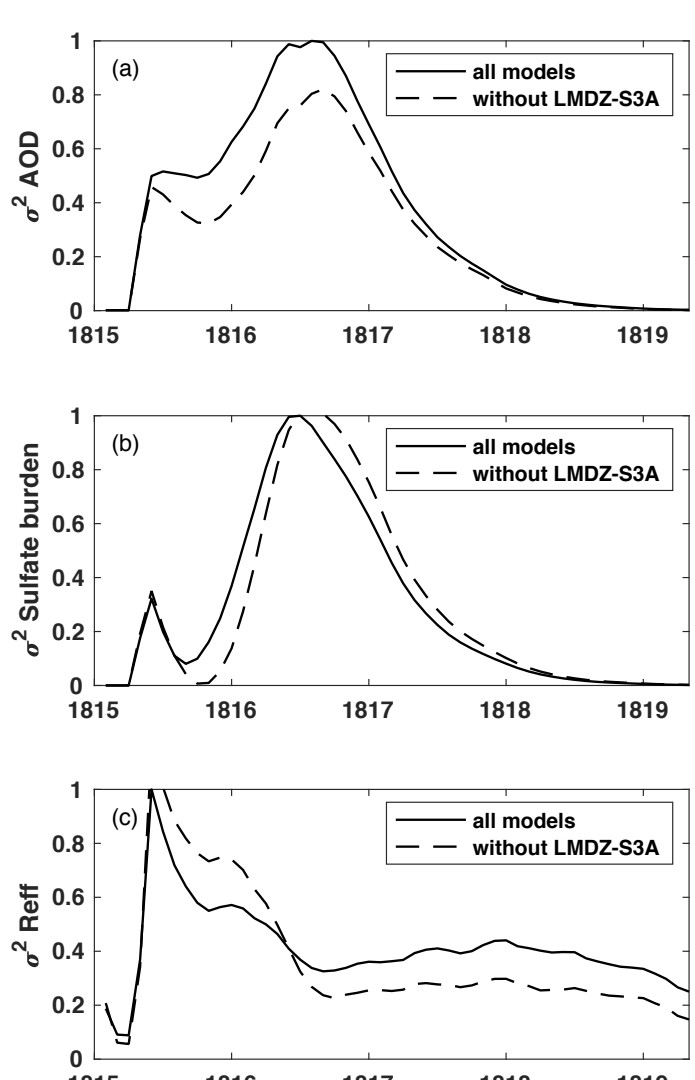


**Figure 7: Variance between VolMIP-Tambora ISA ensemble models for global mean stratospheric a) AOD, b) sulfate burden, and c) effective radius. All models are included in the solid line. All models except for LMDZ-S3A are included in the dashed line. In both cases (solid and dashed lines), the plots have been normalized to the maximum value of the intermodel variance of all models (including LMDZ-S3A) at each corresponding variable. Sulfate burden and effective radius are two of the key output variables dominating the AOD equation, Eq. (1), that generate intermodel variance of AOD.**



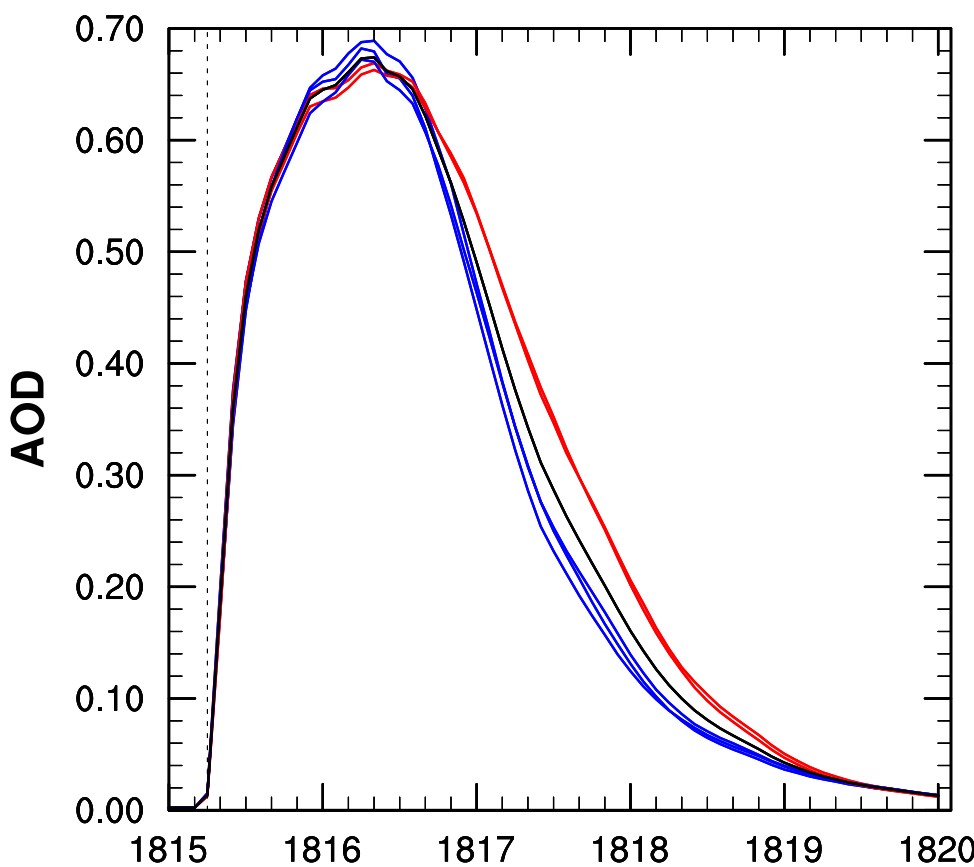

Figure 8: Global stratospheric mean AOD of the 5 CESM-WACCM ensemble runs. Easterly phase nudged QBO forcing is used from the observed strength starting with 1982 for two runs (red), and from the observed strength starting with 1991 for three runs (blue). The ensemble mean of the five runs is in black. Vertical dotted line marks date of injection of $SO_2$.






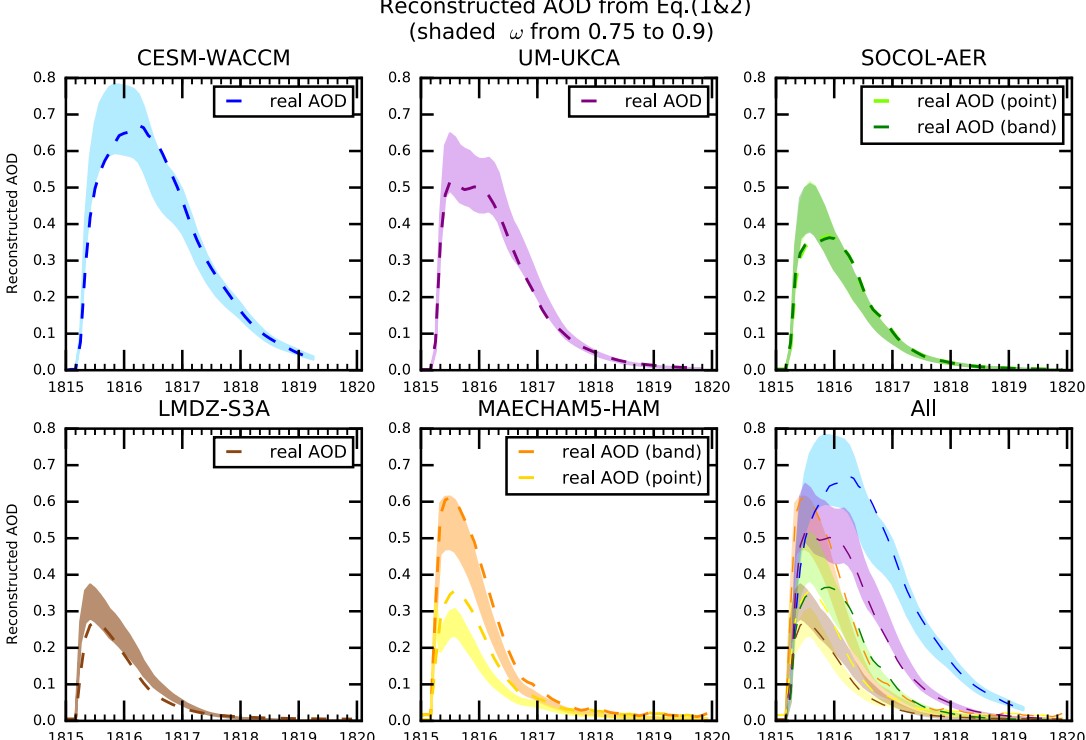

**Figure 9: Reconstructed global stratospheric AOD time series using Eqs. (1 and 2). Shaded regions for each model are from $\omega = 0.9$ (lower edge of shaded region) to 0.75 (upper edge of shaded region). The real AOD from each model is also shown (dashed lines). The dashed lines in this plot are equivalent to the lines in Figure 1. For this plot, the corresponding values of $\rho$ from $\omega$ used for Eq. (2) are calculated using the relationship described by Myhre et al., (2003).**






**Appendix D figures**

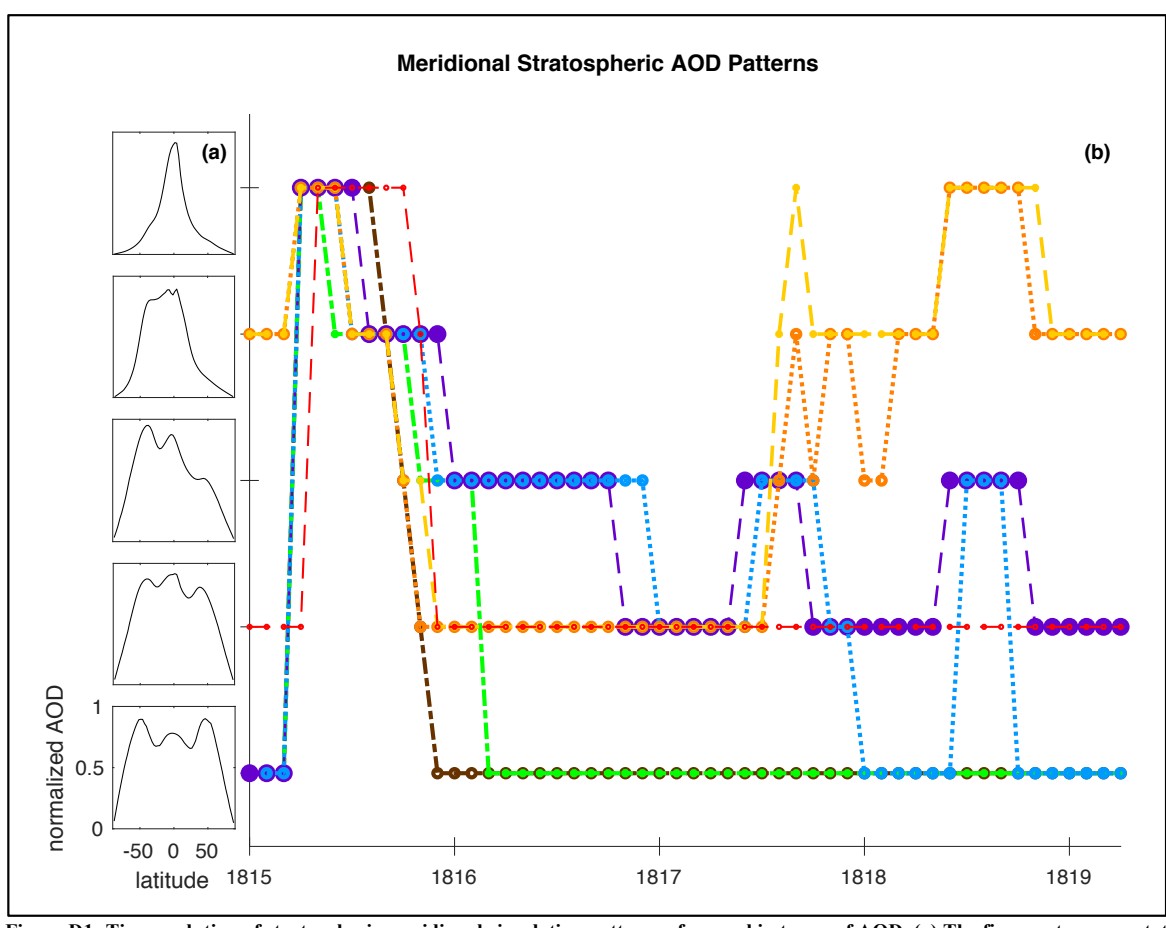

**Figure D1: Time evolution of stratospheric meridional circulation patterns of aerosol in terms of AOD. (a) The five most representative characteristic patterns of normalized zonally averaged AOD as a function of latitude. The five patterns of (a) make up the y-axis of (b), which is the time evolution of each model in terms of which of the five characteristic meridional normalized AOD profiles it best matches at each month. Colors correspond to the ensemble mean of the five ensemble runs from CESM-WACCM (blue), UM-UKCA (purple), SOCOL-AER point (green), MAECHAM5-HAM point (gold), MAECHAM5-HAM band (orange), LMDZ-S3A band (dark brown), and**
**EVA (red).**

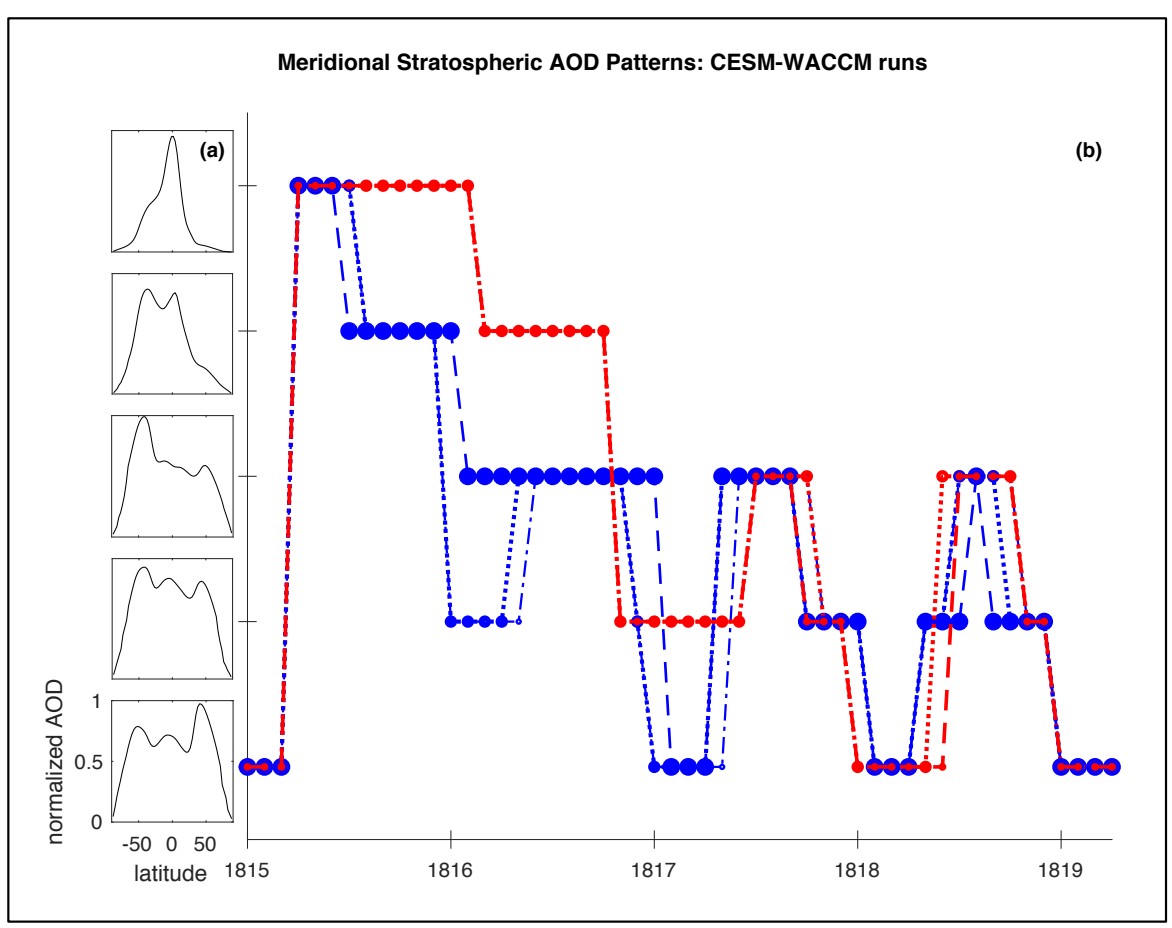

**Figure D2: Time evolution of stratospheric meridional circulation patterns of aerosol in terms of AOD for the five CESM-WACCM ensemble runs. This is presented in the same format as Figure D1 except the meridional AOD patterns in (a) are now derived from only the CESM-WACCM ensemble runs when training the SOM. (b) Time evolution of CESM-WACCM ensemble runs using the easterly QBO forcing observed during 1982 (red) and easterly QBO forcing of 1991 (blue) mapping to which characteristic pattern in (a) best represents the run at each month.**
