# Peer review of "Model physics and chemistry causing intermodel disagreement within the VolMIP-Tambora Interactive Stratospheric Aerosol ensemble"

_Atmospheric Chemistry and Physics, 2020_

## Referee Comment (RC1) · Daniele Visioni (Referee) · 14 Oct 2020

This paper discusses results from preparatory multi-model simulations done as part of the VolMIP project, in particular, simulations of the Tambora 1815 eruption. These results are very useful for understanding where the inter-model differences come from when models are presented with a large stratospheric injection. The authors carefully identify various physical and chemical processes producing those differences, and this represent a great improvement, as the authors point out, compared to the similar Marshall et al. (2018) study, which left some questions unanswered. This paper is therefore extremely interesting and full of information, and it's going to be very useful for the field, so it's a perfect fit for ACP. I have various comments due to the length of the manuscript that I believe would be useful to address to make the manuscript even more useful and clear to the reader.

**Major comments:**

**Introduction**: the introduction dives straight into discussing the VolMIP and ISA-MIP protocols, but gives little to no context. I understand that, for experts in the field, it's irrelevant, but since it's a good introductory paper for the issue of model differences in volcanic eruption simulations, I would suggest some paragraph explaining the motivation for running the experiment, and some further description for the Tambora event.

**ll. 224-227**: I'm not really sure I agree with this conclusion (or maybe I misunderstood it). This is akin to saying that 15% of the $SO_2$ is not converted into stratospheric sulfate aerosol and is instead removed directly. Considering the height of injection, I find this a bit hard to believe, nor do I think the authors have proven this convincingly (and it would be hard to prove, since it would require looking carefully at the 3D strat-trop exchange of $SO_2$, or looking at direct $SO_2$ deposition). Notwithstanding the differences between impulsive injections like this case and sustained injections as in the case of geoengineering simulations, it is not even very consistent with previous findings (see Visioni et al. (2018) for a CCM and a CTM, and Visioni et al. (2020) for CESM(WACCM)) where all $SO_2$ was converted.

Overall, I just don't think the phrase is true, and the fact that (no) "more than 4 TgS is removed before the peak value of global stratospheric sulfate is reached" does not mean that, but rather that some $SO_4$ is removed before the peak (which is to be expected). Unless the authors can prove otherwise.

**l. 237**: A similar note to what I said above. By "Sulfur" what do the authors mean? If they mean aerosols, then it makes sense. If they mean only or also $SO_2$ (as the conclusion to the paragraph above implies) than I don't see any proof of that, but it is an important distinction!

**l. 245**: The phrase "CESM-WACCM and UM-UKCA produce the smallest *Reff*, with values never exceeding 0.5 μm" is a bit in contrast with what previously stated in line 230, where the cause for the increased burden in CESM and UM-UKCA is attributed to the better representation of mid-atmospheric dynamics (assuming that means that the stronger upwelling and confinement in the tropical stratosphere are "correct"). But here it is stated that the two models have the lowest radii, which is a much more straightforward explanation for why the burden is higher. Smaller particles result in less gravitational settling, thus increasing the lifetime (see, for instance, Visioni et al. (2018) Fig. 5 and 7). If the confinement is stronger, particles tend to grow more and thus settle quicker. So overall, the increased burden is much more likely due to these microphysical differences than to dynamical ones (as is also discussed below in Section 4.1).

**l. 444**: I find it a bit puzzling that there is no mention in the list of "Other model uncertainties" of the possible differences in self-lofting resulting from the stratospheric heating produce by the

sulfate aerosols. Together with large-scale dynamical differences discussed in 4.3.2, I would assume that such a large burden would result in large heating rates and thus in an increase in w* compared to unperturbed conditions. Are pre-eruption residual vertical velocities in the tropical pipe comparable between models? And what about heating rates/perturbed w*? It could be that not all models have calculated the transformed eulerian means quantities, which might make this comparison hard to do. If they did (or even if just a few of the models, like CESM and UM-UKCA, for which the authors can't find a clear explanation for the differences) then it would be extremely interesting to show a profile of w* in the tropical pipe. If not, it would be good to at least compare the temperature differences (heating rates are probably more problematic than w*) in the stratosphere and discuss the possibility that the differences are important.

**l. 471 and following**: I'm sorry to say this entire paragraph is utterly confusing to me. I will try to make sense of it, but I suggest a thorough checking by the authors. First, I assume the authors mean "southern" hemisphere. Second, I would say that the injection of $SO_2$ *happens* inside the tropical pipe, it doesn't "go" there. But then, right after, the authors say that the aerosols move "towards the winter pole" which I honestly don't know what it's supposed to mean. And in what sense it "drains" the tropical pipe? I don't understand the following phrase either, about the stratosphere "depth". The tropopause is lower, sure, but I don't get why that would be the explanation for why the AOD moves poleward. And then, see following comment.

**l. 474**: "Aerosols are removed from the high latitudes by tropopause folding" this is a very strong assertion, not proven by the authors for these simulations nor backed up by the literature, as far as I'm aware. Happy to be proven wrong. In general, both for ozone and for aerosols, tropopause folding is *one of the* possible mechanisms, surely not the only one, and I'm not even sure about how predominant it is compared to strat-trop exchange and, for aerosols, gravitational settling. The references I know of (again, mostly for ozone) do not indicate it as the predominant form of contribution to tropospheric air from the tropopause (see for instance Oltmans et al. (1989); Holton et al. (1995); Wimmers et al. (2003); Sprenger et al. 2003)

**Figure D1-D2 and Appendix D**: these figures are very complicated, but that is not the main problem to me. They could be really important, but it feels like the rationale behind using SOM is not explained satisfactorily. Why shouldn't you do a similar analyses using a Dynamical Mode Decomposition instead, or a simple EOF? SOM lets the algorithm decide what the bases are, without a proper physical meaning. These bases are not necessarily orthogonal, so for instance it's not immediate what the differences between the first and the second pattern are. The authors should explain a bit more (feel fry to put it in the supplementary) what SOM is and why it was chosen. This is not something everybody would be familiar with in this field, so it might help people understand what the authors mean.

**Minor comments:**

**l 47**: The ISA acronym needs to be explained here.

**l 49**: I don't really follow what "to effect" means in this context, so the phrase seems a bit obscure to me.

**l 51**: there are word missing. I guess the author wanted to say "since the experiment is designed…"

**l 55**: this part is also a bit hard to understand, and very long. I am only able to understand what HErSEA is because I already know about it, but that "but for the HErSEA experiment" is otherwise a bit obscure (also, the acronym is not explained). I would suggest a stop after 20[th] century, and then a sequent phrase saying: "In most ISA-MIP experiments, the models run…"

**l 56**: I'd remove the word "ensemble" here

**l 58**: "dependence" from initial conditions, or "differences due to" initial conditions

**l 59**: the ISA-MIP protocol prescribed for Pinatubo in Timmreck et al. (2018) a range of SO2 emissions from 10 to 20 Tg-SO2, with a "medium" injection of 14 Tg. The numbers given here result in a low of 14 and a high of 23 Tg. I'd suggest resolving this inconsistency.

**l. 122**: proportional "to"

**l. 209**: I'm not really sure what is intended by "elevated", used here and elsewhere (at least three times in this paragraph). Is it intended as "in the stratosphere"? Or as "high", "large"?

**l. 372**: I don't think "in number" is necessary.

**l. 407**: The version used in Mills et al. (2017) is the same as the one used here, so for consistency it should be called the same (CESM-WACCM)

**l. 476**: please see comments about figure D1.

**l. 509**: This is a very interesting observation, quite in line with a similar effect observed in Visioni et al. (2018).

**l. 607**: "physical and chemical processes"

**l. 625**: "aerosol layer" or "aerosols are spread". "Rather" than a more…

**Figure 4**: hard to read, should be enlarged

**Figure 7**: it would be better if the scales were modified so as to include both lines in all panels. Otherwise the peak for the dashed line can't be evaluated.

**References**

Holton, J. R., Haynes, P. H., McIntyre, M. E., Douglass, A. R., Rood, R. B., and Pfister, L. (1995), Stratosphere-troposphere exchange, *Rev. Geophys.*, 33( 4), 403– 439, doi:10.1029/95RG02097.

Oltmans, S.J., Raatz, W.E. & Komhyr, W.D. On the transfer of stratospheric ozone into the troposphere near the north pole. *J Atmos Chem* 9, 245–253 (1989). https://doi.org/10.1007/BF00052835

Sprenger, M., Croci Maspoli, M., and Wernli, H. (2003), Tropopause folds and cross-tropopause exchange: A global investigation based upon ECMWF analyses for the time period March 2000 to February 2001, *J. Geophys. Res.*, 108, 8518, doi:10.1029/2002JD002587, D12.

Visioni, D., Pitari, G., Tuccella, P., and Curci, G.: Sulfur deposition changes under sulfate geoengineering conditions: quasi-biennial oscillation effects on the transport and lifetime of stratospheric aerosols, Atmos. Chem. Phys., 18, 2787–2808, https://doi.org/10.5194/acp-18-2787-2018, 2018.

Visioni, D., Slessarev, E., MacMartin, D., Mahowald, N. M., Goodale, C. L., and Xia,L.: What goes up must come down: impacts of deposition in a sulfate geoengineering scenario, Environmental Research Letters, 15(9), http://iopscience.iop.org/10.1088/1748-9326/ab94eb

Wimmers, A. J., Moody, J. L., Browell, E. V., Hair, J. W., Grant, W. B., Butler, C. F., Fenn, M. A., Schmidt, C. C., Li, J., and Ridley, B. A. (2003), Signatures of tropopause folding in satellite imagery, *J. Geophys. Res.*, 108, 8360, doi:10.1029/2001JD001358, D4.

---

## Referee Comment (RC2) · Anonymous Referee #2 · 14 Oct 2020

Clyne and co-authors investigate causes of model disagreement in stratospheric aerosol optical depth following the Tambora eruption simulated by the VolMIP models. They show that the differences are largely due to differences in aerosol particle size, and explore the underlying representations of model physics and chemistry to explain the differences. This is no small task, and I congratulate the authors – it's really nice to see such a thorough investigation of inter-model differences. The paper will no doubt serve as a useful point of reference for the VolMIP community and modelling groups. I recommend publication and have made some comments below, most of which are fairly minor – although I do encourage the authors to include uncertainties on Figures 1-3, as discussed below.

[Figure]

Comments:

Line 31: A "pre-study" experiment is unclear – does this refer to an experiment performed before VolMIP officially got underway? Is the experiment not an official VolMIP experiment?

Line 47: define "ISA"

Lines 50-52: Sentence does not quite make sense; I think some words are missing.

Line 118: Please also specify the latitude of Tambora.

Line 280: Section 4.1. Somewhere in this section it would be worth pointing to Appendix A so you can contrast how your 'approximate' AOD was calculated compared with how 'real' AOD was calculated by the models.

Line 677: are there disadvantages to using point eruptions? Otherwise why not recommend that they always be used?

Figures 1-3: I understand that the mean of five ensemble members is shown for each model. It would be good to get some measure of the internal model variation, perhaps by plotting the mean plus/minus one standard deviation in a lighter shade. Otherwise it's hard to assess just how different the models are from one another.

Figure 4: mentions 'Figure 4' twice in the caption.

Figure 9: SOCOL-AER. The green lines are indistinguishable, which might be worth commenting on in the caption.

Please also check the order of figures; it looks like figure 9 is discussed before figure 8.

Appendix D is not referred to in the text, so figures D1 and D2 caught me by surprise – I found myself wondering how the SOMs were created; where the representative patterns had come from, etc. Potentially figures D1 and D2 could be moved to the

main body since they are central to the paper, but I leave that decision to the authors.

---

## Referee Comment (RC3) · Peter Colarco (Referee) · 22 Oct 2020

The paper discusses diversity in the calculated aerosols of several models that simulated a volcanic event based on the 1815 Tambora eruption. The goal of the paper is to understand the reasons for this diversity by looking at specific processes as they are represented in the various models. Simulated stratospheric sulfate mass loading is found to peak at different times in the months following the eruption in the various models, although the total loading reached is generally similar. Peak AOD is found to have similar diversity in the timing in which it is reached, but also the magnitude is found to be very different between the models despite the similar mass loading of

sulfate. Particle effective radius simulated is identified as a key player in the AOD diversity because of sensitivity of extinction efficiency to particle size. Nucleation rate in the LMDZ-S3A aerosol is identified as a significant driver of its outlier performance, but the key difference among the models is between those with interactive OH chemistry (leading to slower conversion of SO2 to sulfate, and so a later peak in AOD) and those without. Simplifying assumptions in some of the models, e.g., relationship of water to optical properties are also highlighted, as are missing processes within the models, including aerosol effects on photolysis and the confounding impact of volcanic ash or water (although since none of the models included either of these it is not a source of discrepancy here but inevitably would be when they start introducing those aspects). For the MAECHAM5-HAM model point versus band emissions of the volcanic injection revealed large differences in both the AOD and effective radius, which is related to its lack of depletion of OH due to using prescribed fields; the SOCAL-AER model (with interactive OH) performed similar injection experiments and did not find significant differences in AOD or effective radius, indicating that in both cases chemistry was slow relative to dynamical transport.

The paper is well written and does not require any major updates. I think it is a useful addition to the community. I suggest mostly minor points to be addressed.

I do agree though with one of the other reviewers that the introduction is a bit wonky going right into the VolMIP protocols and gets kind of technical. I agree that some further context on why any of this is done would help motivate the important results that follow.

Table 1: Please explain/distinguish what is meant for CCM versus AGCM. For example, LMDZ-S3A does not include interactive OH, per Table 2, so why is it a CCM?

Line 131: Requested wavelength of output is 525 nm, but no model provides that, per Table 1 all but SOCOL-AER provide 550 nm. Is there a typo in one place or the other?

It is implied in a few places, but do all the models account for a full suite of troposphericsourced aerosols? Dust, sea salt, carbonaceous, surface SO2 sources? Nitrates too? And to the extent it matters to the results, have you considered interactions of those aerosol sources with the volcanic plume, maybe especially for the modal models? Also, do all the models include an interactive Junge layer aerosol? How about meteoritic smoke?

Line 245 and Appendix A around eqn. A3— This is pedantic, but I got a bit confused following the variable naming and subscripting. Maybe it is all very clear, but it wasn't to me. In A2 I understand reff to the effective radius at a given grid point and moment in time. In A3 Reff is the global averaged effective radius, but curiously Reff appears on both sides of the equation. So I think this is just a typo and "reff" is what is supposed to be on the right side. But A3 pertains to the vertical integration, and the horizontal integration is described somewhat strangely in lines 728-734. I'm not sure if this discussion about gaussian weights is illuminating or necessary consequence of the model output presented on zonal mean grids or what's going on. It might be more clear if A3 were written with the expression wrapped inside of some horizontal integral, is that what's going on?

Line 277: It is interesting to find a statement like "The goal of this paper. . ." in Section 4 of the paper. I think such a clear statement belongs also in Section 1, right before the sentence on line 86 beginning "In this paper we go further. . ."

Line 291: The use of "omega" in a discussion of optical properties could be a bit confusing as it is frequently associated with single scatter albedo. I suggest another symbol. But it's not a big deal, I follow just fine.

Line 300: Speaking of the particles picking up water I presume the hydrated properties are diagnosed and water is not transported on the particle by any of the models. Am I correct?

Figure 5: I was curious what this figure would look like if the circles were sized by effective radius instead of age. Did you make such a plot? I get this is what Figure 6

shows, but on different axes.

Line 359: I note that Figure 9 is introduced before Figure 8, which comes in several pages later.

Line 535: If LMDZ-S3A computed optics assuming omega = 0.75 why does it's internal AOD track so closely with omega = 0.9 curve in Figure 9?

---

## Author Comment (AC1) · 15 Jan 2021

Please see the attached author response to comments (AC1). Thank you!

Please also note the supplement to this comment:
https://acp.copernicus.org/preprints/acp-2020-883/acp-2020-883-AC1-supplement.pdf